# Dynamic simulation of articulated soft robots

Weicheng Huang[1,3], Xiaonan Huang[2,3], Carmel Majidi [2✉] & M. Khalid Jawed [1✉]

Soft robots are primarily composed of soft materials that can allow for mechanically robust maneuvers that are not typically possible with conventional rigid robotic systems. However, owing to the current limitations in simulation, design and control of soft robots often involve a painstaking trial. With the ultimate goal of a computational framework for soft robotic engineering, here we introduce a numerical simulation tool for limbed soft robots that draws inspiration from discrete differential geometry based simulation of slender structures. The simulation incorporates an implicit treatment of the elasticity of the limbs, inelastic collision between a soft body and rigid surface, and unilateral contact and Coulombic friction with an uneven surface. The computational efficiency of the numerical method enables it to run faster than real-time on a desktop processor. Our experiments and simulations show quantitative agreement and indicate the potential role of predictive simulations for soft robot design.

[1] Department of Mechanical and Aerospace Engineering, University of California, Los Angeles, 420 Westwood Plaza, Los Angeles, CA 90095, USA.
[2] Department of Mechanical Engineering, Carnegie Mellon University, 5000 Forbes Avenue, Pittsburgh, PA 15213, USA. [3]These authors contributed equally: Weicheng Huang, Xiaonan Huang. ✉email: cmajidi@andrew.cmu.edu; khalidjm@seas.ucla.edu

Robots composed of soft and elastically deformable materials can be engineered to squeeze through confined spaces[1], sustain large impacts[2], execute rapid and dramatic shape change[3], and exhibit other robust mechanical properties that are often difficult to achieve with more conventional, piece-wise rigid robots[4–10]. These platforms not only exhibit unique and versatile mobility for applications in biologically inspired field robotics, but can also serve as a testbed for understanding the locomotion of soft biological organisms. However, owing to the current limitations with simulating the dynamics of soft material systems, design and control of soft robots often involve a painstaking trial and error process, and it can be difficult to relate qualitative observations to underlying principles of kinematics, mechanics, and tribology. Progress, therefore, depends on a computational framework for deterministic soft robot modeling that can aid in design, control, and experimental analysis.

Previous efforts to simulate soft robots have focused on Finite Element Method[11–16], voxel-based discretization[17,18], and modeling of slender soft robot appendages using Cosserat rod theory[19–21]. Drawing inspiration from simulation techniques based on discrete differential geometry (DDG) that are widely used in the computer graphics community[22], we introduce a DDG-based numerical simulation tool for examining the locomotion of limbed soft robots. The DDG approach starts with discretization of the smooth system into a mass-spring-type system, while preserving the key geometric properties of actual physical objects; this type of simulation tool is naturally suited to account for contact and collision[23]. In particular, we treat the robot as being composed of multiple slender actuators that can be modeled using elastic rod theories[24–28]. In order to achieve rapid simulation runtimes, we adapt fast and efficient physically based computational techniques that have gained traction within the computer graphics community to model slender structures, e.g., rods[29–31], ribbons[32], plates[33], shells[34], viscous threads[30,35], and viscous sheets[36]. Despite the visual realism in these simulation methods, these prior works do not comprehensively capture all the physical ingredients for a physically accurate simulation of fast moving articulated soft robots. Our numerical method integrates these ingredients—frictional contact, material damping, and inertial effects—into a discrete simulation framework to achieve quantitative agreement with experiments. Recently, a DDG-based formulation was used to model a caterpillar-inspired soft robot in which the individual segments of the robot were treated as curved elastic rod elements[37]. Although promising, this formulation could not accurately capture inertial effects—a key feature of fast moving robots—and did not incorporate the necessary contact and friction laws required to achieve quantitative agreement with experimental measurements.

Here, we employ a discrete representation of a soft robot and incorporate Coulomb frictional contact, inelastic collision with ground, and inertial effects in a physically accurate manner. The mechanical deformation of the robot is associated with local elastic (stretching and bending) energies at each discrete node. We formulate these discrete elastic energies and, subsequently, the discrete equations of motion representing the balance of forces using principles from classical elastic rod theories[29,38]. Coulomb frictional contact with uneven surface is integrated into the formulation using the modified mass method[33], such that a group of constrained equations of motion can be implicitly updated through a second order, symplectic Newmark-beta time integration scheme. As this integration scheme is momentum preserving, it does not suffer from artificial energy loss—a well-known attribute of first order Euler integration used in prior works with discrete rod simulations[29]—and can capture the essential inertial effects during the dynamic simulation of soft robots. The elastic/ inelastic collision between the soft robot and rigid ground can be

captured by the rate-dependent viscoelastic behavior of the soft material, i.e., the damping coefficient in Rayleigh's damping matrix is used to precisely control the recovery factor during collision and rebound[39]. Finally, the experimentally measured data of a single actuator during one actuating–cooling cycle is fed into our numerical framework for the investigation of soft robotic dynamics. The result is a robust simulation tool that can run faster than real-time on a single thread of a desktop processor. The reliability of this simulation tool for making quantitative predictions is systematically examined using three test cases. First, we demonstrate that three empirically observed motion patterns of a deformable rolling ribbon[40] on a declined surface can be captured by our simulator. Next, we build two types of soft robots made of SMA-based limb: a star-shaped rolling robot composed of seven radially oriented limbs and a jumper robot with a single limb. The SMA-based robots were selected because of the ability to achieve rapid dynamic motions in which both material deformation and inertia have a governing role[41,42]. In order to examine the influence of friction and ground topology, locomotion experiments were performed on flat, inclined/declined, and wavy/undulating surfaces. In all cases, we found reasonable quantitative agreement between experiments and simulations.

## Results

**Numerical simulation.** In this section, we review the numerical framework that incorporates elasticity, contact with uneven surface, friction, and inelastic collision for a comprehensive soft robot simulator. As the motion of the robot remains in 2D, we do not include a twisting energy of the rod, although this can be readily integrated into our framework[29]. Starting from the discrete representation of elastic energies, we formulate equations of motion at each node and update the configuration of the structure (i.e., position of the nodes) in time. The rod segment between two consecutive nodes is an edge that can stretch as the robot deforms—analogous to a linear spring. The turning angle $\phi_i$ (see Fig. 1b) at node $x_i$ between two consecutive edges can change— similar to a torsional spring. The elastic energy from the strains in the robot can be represented by the linear sum of two components: stretching energy of each edge and bending energy associated with variation in the turning angle at the nodes. The discrete stretching energy at the edge connecting $x_i$ and $x_{i+1}$

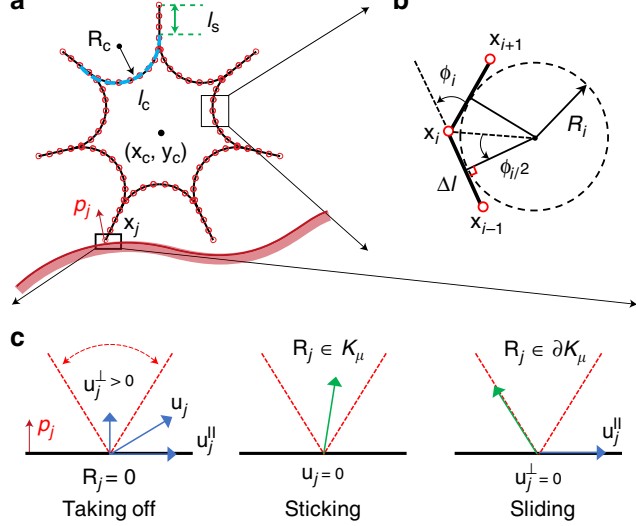

**Fig. 1 Discrete kinematic representation of a soft robot. a** Geometric discretization of soft rolling robot. **b** The bending curvature at $i$th node is $\kappa_i = 1/R_i = 2\tan(\phi_i/2)/\Delta l$[30]. **c** Coulomb law for frictional contact[45].

is $E_i^s = \frac{1}{2}EA\epsilon^2$, where $EA$ is the stretching stiffness (calculated as the product of the material elastic modulus $E$ and actuator cross-sectional area $A$) and $\varepsilon_i = |\mathbf{x}_{i+1} - \mathbf{x}_i|/\Delta l - 1$ is the axial stretch. Associated with each turning angle $\phi_i$ is the discrete bending energy $E_i^b = \frac{1}{2}EI(\kappa_i - \bar{\kappa}_i)^2\Delta l$, where $EI$ is the bending stiffness, $\kappa_i = 2\tan(\phi_i/2)/\Delta l$ is the curvature [Fig. 1b], and $\bar{\kappa}_i$ is the natural curvature (i.e., curvature evaluated in undeformed configuration). In the special case of a joint node where three edges meet, the bending energy is comprised of two components: one corresponding to the turning angle between the first and second edges and the second one arises from the turning angle between the second and third edges. The total stretching energy of the robot can be obtained simply by summing over all the edges, i.e., $E^s = \sum_i E_i^s$, and, similarly, the total bending energy is $E^b = \sum_i E_i^b$. In both experiments and simulations, we observe that the structure is nearly inextensible and the prominent mode of deformation is bending. We evaluated the bending stiffness by quantifying the shape of an actuator under vertical load, as shown in Supplementary Fig. 1 and Supplementary Methods.

The elastic stretching (and bending) forces acting on a node $\mathbf{x}_i$ can be obtained from the gradient of the energies, i.e., $-\left[\frac{\partial E^s}{\partial x_i}, \frac{\partial E^s}{\partial y_i}\right]^T$ (and $-\left[\frac{\partial E^b}{\partial x_i}, \frac{\partial E^b}{\partial y_i}\right]^T$). An implicit treatment of the elastic forces requires calculation of the $2N \times 2N$ Hessian matrix of the elastic energies. Other than the seven joint nodes that are connected with three other nodes, a node $\mathbf{x}_i$ is only coupled with the adjacent nodes $\mathbf{x}_{i-1}$ and $\mathbf{x}_{i+1}$ in the discrete energy formulation. This results in a banded Hessian matrix with $6 \times 6$ blocks of non-zero entries along the diagonal. The only off-diagonal non-zero entries correspond to the seven joint nodes. The analytical expressions for the gradient and Hessian of the elastic energies can be found in refs. [29,30].

Besides the internal elastic forces, $\mathbf{F}^s$ and $\mathbf{F}^b$, the structure would also experience internal damping forces during deformation. We use the Rayleigh damping matrix to formulate the viscoelastic behavior of soft robots, such that the damping force vector is given by[39]

$$\mathbf{F}^d = -(\alpha\mathbb{M} + \beta\mathbb{K})\mathbf{v}, \qquad (1)$$

where $\alpha, \beta \in \mathbb{R}^+$ are damping coefficients, $\mathbb{K} = -\frac{\partial}{\partial \mathbf{q}}\left(\mathbf{F}^s + \mathbf{F}^b\right)$ is the tangent stiffness matrix, and $\mathbf{v}$ is the velocity vector (time derivative of DOF (degree of freedom)). Also, the external gravity forces are denoted by $\mathbf{F}^g$, as well as the external contact forces, $\mathbf{F}^r$. The gradients of these force vectors can be analytically formulated in a manner similar to those of the elastic forces. The sparse nature of the Jacobian matrix is critical for computational efficiency during the solution of the equations of motion, described next.

The DOF vector can be updated from current time step ($t_k$) to the next ($t_{k+1} = t_k + h$), $\mathbf{q}^{k+1} = \mathbf{q}^k + \Delta\mathbf{q}^{k+1}$, by a second order, implicit Newmark-beta time integration[39],

$$
\begin{aligned}
\Delta\mathbf{q}^{k+1} - h\mathbf{v}^k &= \frac{h^2}{4}\mathbb{M}^{-1}\left(\mathbf{F}^{k+1} + \mathbf{F}^k\right) \\
\Delta\mathbf{q}^{k+1} &= \frac{h}{2}\left(\mathbf{v}^{k+1} + \mathbf{v}^k\right) \\
\Delta\mathbf{v}^{k+1} &= \mathbf{v}^{k+1} - \mathbf{v}^k,
\end{aligned}
\qquad (2)
$$

where the velocity vector (time derivative of DOF) is $\mathbf{v}$, superscript $k+1$ (and $k$) denotes evaluation of the quantity at time $t_{k+1}$ (and $t_k$), $\mathbb{M}$ is the diagonal mass matrix, $h$ is the time step size, and $\mathbf{F} = (\mathbf{F}^s + \mathbf{F}^b + \mathbf{F}^g + \mathbf{F}^d + \mathbf{F}^r)$ is the sum of elastic, damping, and external forces defined before. In the absence of dissipative forces and external contact forces, this method is symplectic and momentum preserving[39,43,44]—a critical feature for simulation of robots where inertial effects are significant.

As soft robots are often intended for locomotion on unstructured terrain, we require a method to account for contact and friction with the ground. Importantly, the surface normal can vary with the horizontal $x$ axis. We model the nonpenetration constraints and frictional contact forces that resist sliding along interfaces based on Coulomb's law. At each time step, we apply continuous collision detection to the predicted trajectory to gather contact constraints into a contact set $\mathbb{C}$, shown in Fig. 1c. For these calculations, the velocity $\mathbf{u} = [\mathbf{v}_{2j-1}, \mathbf{v}_{2j}]^T$ (subscript denotes element number in a vector), and the reaction force $\mathbf{R} = [\mathbf{F}_{2j-1}^r, \mathbf{F}_{2j}^r]^T$, at the $j$th node (the contact point) satisfy the condition

$$\mathbb{C}(\mathbf{u}, \mathbf{R}) \Leftrightarrow \begin{cases} \mathbf{R} = 0 \text{ and } u^\perp > 0 \text{ (take off)} \\ R_\| < \mu R^\perp \text{ and } \mathbf{u} = 0 \text{ (sticking)} \\ R_\| = \mu R^\perp \text{ and } u^\perp = 0 \text{ (sliding)}, \end{cases} \qquad (3)$$

where $\mu = 0.8$ is the friction coefficient characterized by experiments (Supplementary Methods), and the superscript $\|$ (and $\perp$) denotes the component along (and perpendicular to) the ground. At the normal and tangential subspaces of a contact node $\mathbf{x}_j$, we either know its perpendicular velocity $u^\perp$ ($u^\|$ for tangential component) or the perpendicular reaction force $R^\perp$ ($R^\|$ for tangential component), so the Coulombic frictional contact law can be treated as a Second Order Linear Complementary Problem (SOLCP)[45]. We employ the modified mass method[33] to solve this SOLCP such that a contact node $\mathbf{x}_j$ can be free (degrees of freedom is 2, taking off), constrained along the normal to the ground $\mathbf{p}$ (degrees of freedom is 1, sliding), or fully constrained (degrees of freedom is 0, sticking). The two modified equations of motion for the $j$th node ($j = 1, ..., N$) are

$$\begin{bmatrix} \mathbb{F}_{2j-1} \\ \mathbb{F}_{2j} \end{bmatrix} \equiv \begin{bmatrix} \Delta\mathbf{v}_{2j-1}^{k+1} \\ \Delta\mathbf{v}_{2j}^{k+1} \end{bmatrix} - \frac{h}{2M_j}\mathbf{S}^{k+1}\left( \begin{bmatrix} \mathbf{F}_{2j-1}^{k+1} \\ \mathbf{F}_{2j}^{k+1} \end{bmatrix} + \begin{bmatrix} \mathbf{F}_{2j-1}^k \\ \mathbf{F}_{2j}^k \end{bmatrix} \right) - \Delta\mathbf{z}^{k+1} = 0, \qquad (4)$$

where $\mathbb{F}_{2j-1}$ is the left hand side of the $(2j-1)$-th equation of motion, $M_j$ is the mass associated with $j$th node, $\Delta\mathbf{z}^{k+1}$ is the change in velocity we want to enforce along the constrained direction(s), and the modified mass matrix is

$$\mathbf{S}^{k+1} = \begin{cases} \mathbb{I} & \text{if ndof} = 2, \\ (\mathbb{I} - \mathbf{p}\mathbf{p}^T) & \text{if ndof} = 1, \\ 0 & \text{if ndof} = 0, \end{cases} \qquad (5)$$

where ndof is the number of free DOF at $j$th node and $\mathbb{I}$ is the $2 \times 2$ identity matrix. Note that when a node is free, $\Delta\mathbf{z}^{k+1} = \mathbf{0}$, and Equation (4) reduces to Equation (2). If the node is fully constrained ($\mathbf{S}^{k+1} = \mathbf{0}$), Equation (4) reduces to $\Delta\mathbf{v}_j^{k+1} = \Delta\mathbf{z}^{k+1}$ and the change in velocity (as well as the position) is enforced to take the value prescribed by $\Delta\mathbf{z}^{k+1}$.

The solution to the $2N$ equations of motion in Equation 4 starts with an initial guess $(\Delta\mathbf{v}^{k+1})^{(0)}$ and subsequent Newtons iterations to improve the solution until a desired tolerance is achieved:

$$\left(\Delta\mathbf{v}^{k+1}\right)^{(n+1)} = \left(\Delta\mathbf{v}^{k+1}\right)^{(n)} - \mathbb{J}^{(n)}\backslash\mathbb{F}^{(n)}, \qquad (6)$$

where $\mathbb{J}^{(n)} = \frac{\partial\mathbb{F}}{\partial(\Delta\mathbf{v}^{k+1})}$ is the Jacobian matrix evaluated at $(\Delta\mathbf{v}^{k+1})^{(n)}$. The non-trivial terms in the evaluation of this Jacobian are the Hessian matrices of the elastic energies. Owing to the presence of the ground, we need to check whether the new solutions, e.g.,

$\mathbf{q}^{k+1}$, $\mathbf{v}^{k+1}$, and $(\mathbf{F}^r)^{k+1}$ (computed from force balance), satisfy the following conditions:

- A node $\mathbf{x}_j$ cannot fall below the ground.
- The normal component of reaction force $R^\perp$ exerted by the ground on a node $\mathbf{x}_j$ must be along the outward normal to the surface, e.g., $R^\perp > 0$.
- The reaction force $\mathbf{R}$ should be in the frictional cone zone $K_\mu$ (see Fig. 1c); if the reaction force is on the boundary of the cone, this node is allowed to slide along the tangential direction of surface opposite to reaction force, $\mathbf{u} \cdot \mathbf{R} < 0$.
- If the tangential velocity $u^\parallel$ at a sliding node $\mathbf{x}_j$ changes its direction, $(u^\parallel)^k \cdot (u^\parallel)^{k+1} < 0$, this node should be fully constrained.

If one of the above rules is broken, we rewind the simulation, add (or delete) constraints at the contact pair, and re-solve Equation (4) with a new guess.

When an elastic body drops onto a rigid surface, the motion normal to the surface of the contact nodes are constrained, the normal velocities are set to zero, and the tangential velocities are reduced based on impulse theory, $\Delta u^\parallel = \mu \Delta u^\perp$. If the structure is modeled as an ideal mass-spring system without viscoelasticity, the whole structure will rebound to a certain height and the recovery factor—the ratio of rebound to initial height—is not deterministic. This arises because the structure's kinetic energy will transfer into elastic potential energy during compression and then convert back to kinetic energy during the rebound phase[39]. We must account for the rate-dependent viscoelasticity of contact for predictive simulation, where the energy loss of the collision–compression–rebound process results in a deterministic rebound height. In Supplementary Fig. 3 and Supplementary Methods, we show that the decrease in rebound height of the rolling robot can be determined by the parameter $\beta$ in damping force $\mathbf{F}^d = -(\alpha \mathbb{M} + \beta \mathbb{K})\mathbf{v}$, such that the recovery factor of collision is also related to $\beta$. Physically, $\beta$ represents a damping that opposes elastic deformation, without penalizing rigid body motion. Opposition to rigid body motion and momentum dissipation can be accounted by the viscosity $\alpha$.

The overall numerical framework thus accounts for inertia, friction, and collision and shows good convergence with both time and space discretization, as outlined in Supplementary Figs. 4, 5, and Supplementary Methods.

**Rolling ribbon.** Before examining soft robot locomotion, we first investigate the simpler motion of a circular ribbon on a declined surface in order to test the accuracy of numerical implementation of friction and contact. In the numerical study here, the arc length we chose for the circular ribbon is $L_0 = 0.3$ m, resulting in $R = 0.3/2\pi \approx 0.048$ m (details in Supplementary Fig. 7 and Supplementary Methods). Because of gravity, this close-loop elastic structure will first undergo transient dynamics and then, as shown in Fig. 2a, move with a steady state configuration. The final shape is determined by the ratio $\Gamma_g = L_g/R$ of the gravito-bending length scale $L_g = (EI/\rho g A)^{1/3}$ to the ribbon undeformed radius $R$[40]. In Fig. 2c, we plot the static configurations of rolling ribbon at different values of $\Gamma_g$. At small values of $\Gamma_g$, the ribbon shows relatively large deformation with large region of contact. As $\Gamma_g$ increases, the deformed shape becomes closer to its original undeformed shape and the contact length decreases to reach a single point at $\Gamma_g = \infty$.

Now we turn to the motion of a rolling ribbon. Three different motion patterns exist on a declined surface: pure sliding, combined sliding and rotation, and pure rotation, depending on a dimensionless number, $\mu/\tan \theta$, where $\mu$ is the frictional coefficient and $\theta$ is the decline angle. In Fig. 2b, we show the ratio between the distance traveled by a point on the ribbon (red mark in Fig. 2a) and the ribbon centroid, $\delta$, as a function of normalized friction coefficient, $\mu/\tan \theta$, at different values of $\Gamma_g$. When the normalized frictional coefficient $\mu/\tan \theta = 0$, the ribbon will slide along the tangential direction of the surface without any

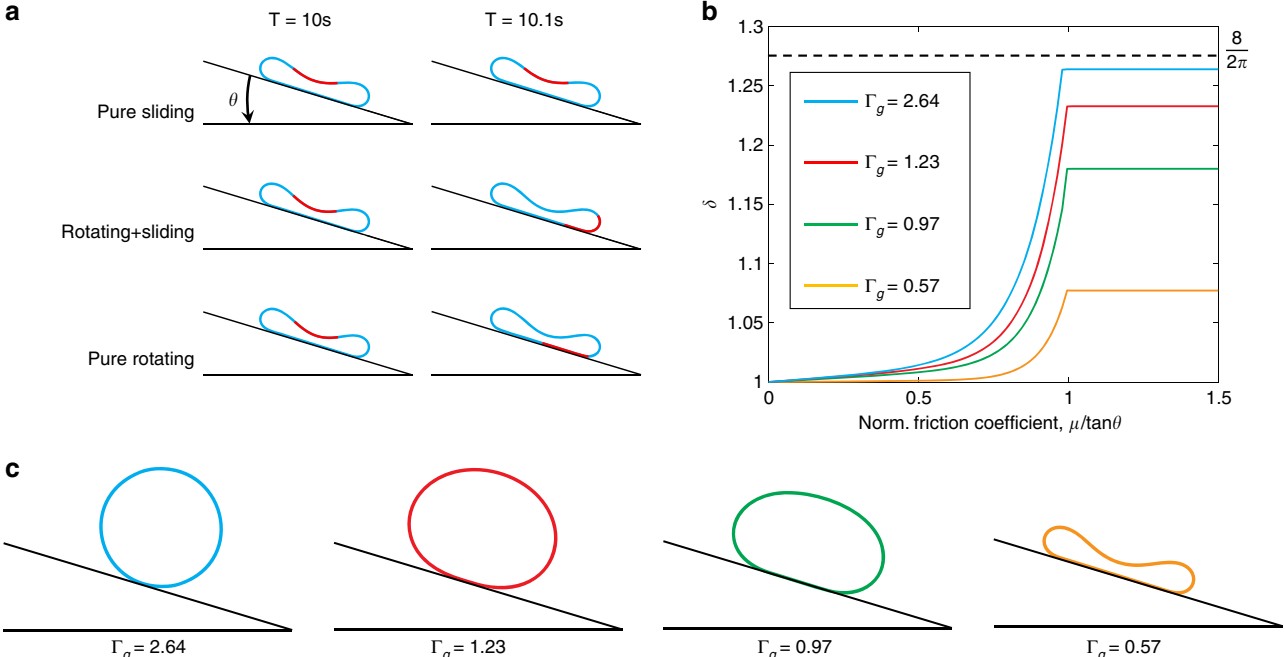

**Fig. 2 Motion patterns of elastic ribbons. a** Three different patterns in rolling ribbons: pure sliding ($\mu/\tan \theta = 0$), combination of sliding and rotating ($0 < \mu/\tan \theta < 1$); and pure rotating ($\mu/\tan \theta \geq 1$). **b** The ratio between the route of ribbon boundary point and ribbon centroid, $\delta = s_b/s_c$, as a function of normalized frictional coefficient $\mu/\tan \theta$, for different values of normalized ribbon curvature, $\Gamma_g$. **c** Different typologies of rolling ribbons with different $\Gamma_g$.

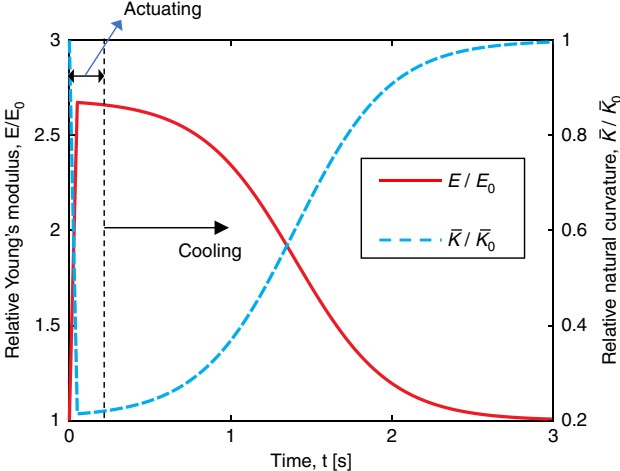

**Fig. 3 Actuator characterization.** Relative natural curvature $\bar{\kappa}/\bar{\kappa}_0$ and Young's modulus $E/E_0$ as a function of time during actuating–cooling process of a single SMA actuator.

rotation, and the path of boundary point is the same as the path of center, $\delta = 1$. If $0 < \mu/\tan\theta < 1$, the motion of the ribbon is a combination of sliding and rotation, and the larger the friction, the higher the $\delta$. The ribbon undergoes pure rotation at $\mu/\tan\theta \geq 1$ when $\delta$ remains fixed at a constant value depending on $\Gamma_g$. At the limiting case of a rigid ribbon, the motion is purely rotational and any point on the ribbon traces a cycloid path, corresponding to $\delta = 8/2\pi$. We plot the boundary node position as a function of time for all three cases in the Supplementary Fig. 6 to better show their differences. This finding establishes that the simulation can systematically capture elasticity, friction, and their interplay.

**Rolling robot**. The star-shaped, rolling robot in Fig. 1a is comprised of seven compliant actuators/limbs that are arranged radially. Each limb has a curved part with length $l_c = 2.2$ cm and a straight part with length $l_s = 0.8$ cm. The natural curvature of the curved part is $\bar{\kappa}_0 \equiv 1/R_c = 120$ m$^{-1}$. The material density of the rolling robot is $\rho = 1912$ kg m$^{-3}$. The mass center is located at $(x_c, y_c)$. The height, $H \approx 5$ cm, is used as the body length. We then discretize the structure into $N$ nodes, shown schematically in Fig. 1a. This corresponds to a DOF vector, $\mathbf{q} = [x_0, y_0, ..., x_{N-1}, y_{N-1}]^T$, of size $2N$, representing the vertical and horizontal coordinates of each node. Here, the superscript $T$ denotes transposition. The length of each edge—the segment between two consecutive nodes—in this study is $\Delta l \approx 2.5$ mm, resulting in $N = 84$ nodes (convergence study in Supplementary Fig. 4).

Actuation is incorporated into the simulation by varying natural curvature and bending stiffness with time. This variation is measured through characterization of a single SMA-powered actuator, as described next. The electrically activated SMA wire enables rapid transition between a soft curled unactuated state and a stiff straight-like actuated state[41,42]. The relative natural curvature $\bar{\kappa}/\bar{\kappa}_0$ and Young's modulus $E/E_0$ are temperature-dependent, and can change as a function of time during the actuating–cooling process. As shown in Fig. 3, when SMA is actuated for 0.25 s, its natural curvature and Young's modulus increase linearly in a short time period, $t_0$, followed by a logistic decay until reaching the unactuated state. Notice that the plot here is from experimental fitting, see Supplementary Fig. 2 and Supplementary Methods for details. We use a piece-wise function

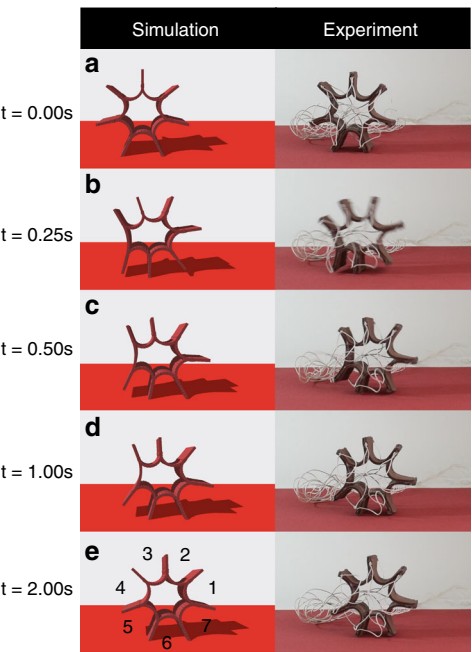

**Fig. 4 Snapshots of rolling robot. a–e** Snapshots of rolling robot from simulations and experiments between $t = 0.0$ s and $t = 2.0$ s. Limb #4 is actuated for a rolling motion. The width (out-of-plane dimension in this figure) is 18 mm.

to describe the natural curvature of SMA actuators:

$$\bar{\kappa}(t) = \begin{cases} \frac{(n-1)t}{t_0}\bar{\kappa}_0 + \bar{\kappa}_0 & \text{when } t < t_0 \\ \frac{(1-n)}{1+e^{-\tau(\bar{t}-t)}}\bar{\kappa}_0 + n\bar{\kappa}_0 & \text{when } t > t_0, \end{cases} \quad (7)$$

where $n = \bar{\kappa}_{\min}/\bar{\kappa}_0$ is the ratio between the minimum curvature (at $t = t_0$) and the initial curvature (at $t = 0$), and $\tau, \bar{t}, t_0$ are numerical parameters obtained from experimental fitting. The change of Young's modulus of SMA follows a similar piece-wise function. Note that the parameter $t_0$ is not necessarily equal to the actuation time of 0.25 s. As a result, the curvature slightly increases (and Youngs modulus decreases) even when the actuator is being heated (at $t_0 < t < 0.25$ s). The primary reason behind behavior is that the fitting function is constrained to be smooth and monotonic (i.e., either increase, decrease, or remain constant). Although we could separate the fitting into more than two piece-wise functions involving more fitting parameters, this will lead to added complexity with little improvement in fitting accuracy. With these fitting parameters, we can achieve excellent match between experimental measurements and numerical simulations performed on a single actuator (details on the fitting can be found in Supplementary Fig. 2 and Supplementary Methods).

The simplest scenario is presented in Fig. 4 (also Supplementary Movie 1), where the surface normal is anti-parallel to gravity. Figure 5d plots the $x$ coordinate of the centroid of the robot, $x_c$, with time over four actuation cycles. Note that the different symbols correspond to repeated experimental runs. In our numbering system for the limbs (see Fig. 4), Limb 5 is in contact with the ground at $t = 0$. Upon actuation of Limb 4, the robot rolls to the right and the contact limb changes from 5 to 6. In the next cycle, Limb 5 (the limb to the left of the contact limb) is actuated. We choose the actuation period $\Delta t = 3$ s (0.25 s for actuation and 2.75 s for cooling); the single SMA actuator can totally reshape to its original configuration within 3 s.

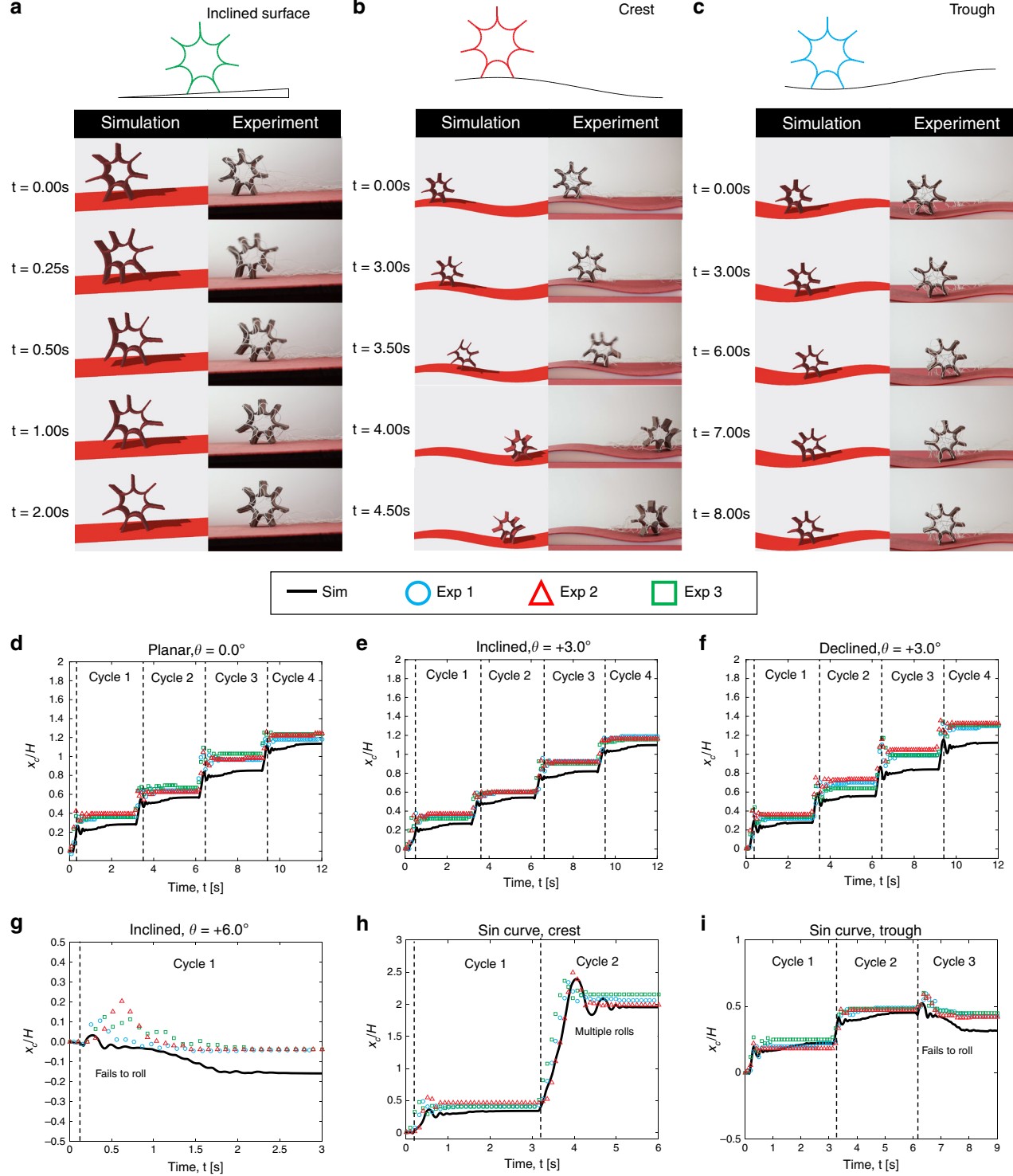

**Fig. 5 Snapshots of rolling robot in different scenarios. a** Inclined surface ($\theta = +3.0°$), **b** Crest of sinusoidal surface, and **c** Trough of sinusoidal surface. Normalized $x$ coordinate of the robot centroid with time from experiments (symbols) and simulations (solid lines): **d** planar surface, $\theta = 0.0°$; **e** inclined surface, $\theta = +3.0°$; **f** declined surface, $\theta = -3.0°$; **g** inclined surface, $\theta = +6.0°$; **h** sinusoidal surface, crest; and **i** sinusoidal surface, trough.

Next, we consider planar surfaces that are inclined at an angle $\theta$ with respect to the horizontal plane (also see Supplementary Movie 1). Figure 5a compares the simulation and experimental results for $\theta = +3.0°$, and Fig. 5e–g plots the location of the robot centroid at three different values of $\theta$. We find good agreement between experiments and simulation in all the cases. In particular, we observe that when the angle of inclination increases from $\theta = -3.0°$ to $\theta = +3.0°$, the distance traveled by the robot decreases in

both experiments and simulations. The gait at $\theta = \{-3.0°, +3.0°\}$ is similar to the horizontal planar case described above. Beyond a certain threshold for $\theta$, the robot can no longer move forward owing to the increased role of gravity, e.g., the robot fails to roll up the incline when at $\theta = +6.0°$. The simulation also accurately captures this observation.

We now move to the case of an uneven surface with an outward normal that varies with location. As a representative

example shown schematically in Fig. 5b, c, we consider a 3D printed surface that can be described by $f(x) = A \sin(2\pi x/\lambda)$ with amplitude $A = 6.5$ mm and period $\lambda = 200$ mm. We consider two experimental trials: first, the robot is initially located at the crest of the surface in Fig. 5b; and second, the robot is on the trough in Fig. 5c. Figure 5h, i shows the location of the robot centroid with time from both experiments and simulations. In the crest case, the robot rolls once at the first cycle. However, at the second cycle, the robot rolls multiple times, undergoes oscillatory motion, and settles stay at the trough. On the other hand, if the locomotion starts with the robot at the trough, the robot successfully rolls once in the first two cycles, but fails to roll in the third cycle. All of these observations are captured in both experiments and simulations. However, we should also note that our simulator always under-predicts the motion of the rolling robot. We attribute this to the finite thickness of the actuator elements, which is not accounted for in the model.

Our novel numerical tool can achieve real-time simulation of the soft rolling robot. In Fig. 6, with a fixed number of vertices, $N = 84$, the computation time linearly scales with time step size $h$ for all the scenarios. The simulations ran on a single thread of AMD Ryzen 1950X CPU @ 3.4 GHz. Also, our simulator can run faster than real time when the time step size $h \gtrsim 2.5$ ms. Numerical issues associated with a large step size appear at $h \gtrsim 10$ ms, in which case the computation time is infinite because we cannot get convergence.

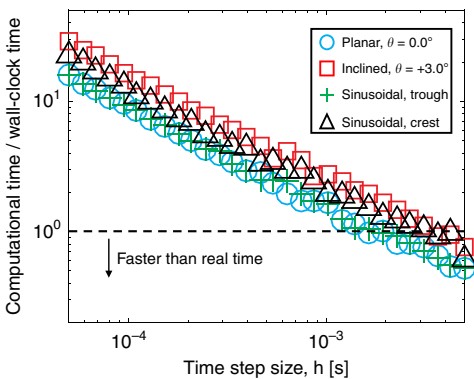

**Fig. 6 Computational time.** The ratio between computational time and wall-clock time as a function of time step size $h$ in different scenarios with $N = 84$.

**Jumper.** Finally, we emphasize the generality of the simulation by examining another soft robot with a different geometry. The SMA-based jumper shown in Fig. 7a is an asymmetric circle with radius $\bar{R}_0 \approx 5$ mm. Detailed geometric property of Jumper robot can be found in the Supplementary Fig. 7 and Supplementary Methods. When the material is actuated, the whole structure can rise and move forward because of the reaction forces from the ground. To model the tension from the electrical wire connected at the leading edge of the jumper, we apply a force at the first node; the magnitude and duration of the force are obtained from fitting to experimental data (Supplementary Fig. 7). In Fig. 7a, we show snapshots of the jumper at $t = \{0.000, 0.125, 0.250\}$s from both experiments and simulations and see qualitative agreement. For quantitative comparison, Fig. 7b, c present experimental and simulation data on the normalized position of the first node on the robot as a function of time. The two sets of results—experiments and simulations—appear to be in strong quantitative agreement, providing further evidence for the physical accuracy of our DDG-based formulation. We should also note that, at time $t = 1.5$ s, the $x$ position predicted by numerical simulation has a sudden drop, which is not reflected in the experimental data. The mismatch between numerical simulation and experimental observation is owing to the discontinuity in Coulombs formulation, i.e., the jumper robot would move only if the external reaction force is out of the frictional cone.

## Discussion

We have introduced a numerical framework based on DER for examining the locomotion of limbed soft robots that is adapted from methods popular in the computer graphics community. To avoid the artificial energy dissipation during the time marching scheme, we replaced the first order, implicit Euler integration by a second order, symplectic Newmark-beta method, for momentum preservation during the dynamic simulation. For the frictional contact between the rigid wall and the soft material, Coulomb's law was implemented through a modified mass-based method, and this fully implicit framework allows a larger time step for convergence and numerical stability, which is also a prerequisite for real-time simulation. Similarly, the elastic/inelastic collision between the rigid wall and soft robots, related to the rate-dependent viscoelastic behavior of soft material, can be precisely described by the Rayleigh damping matrix. The mechanical response of SMA during actuating–cooling process was first experimentally measured through a single actuator, then fed to the numerical framework to simulate the dynamics of the soft robots. Overall, the simulation can seamless integrate elasticity, actuation,

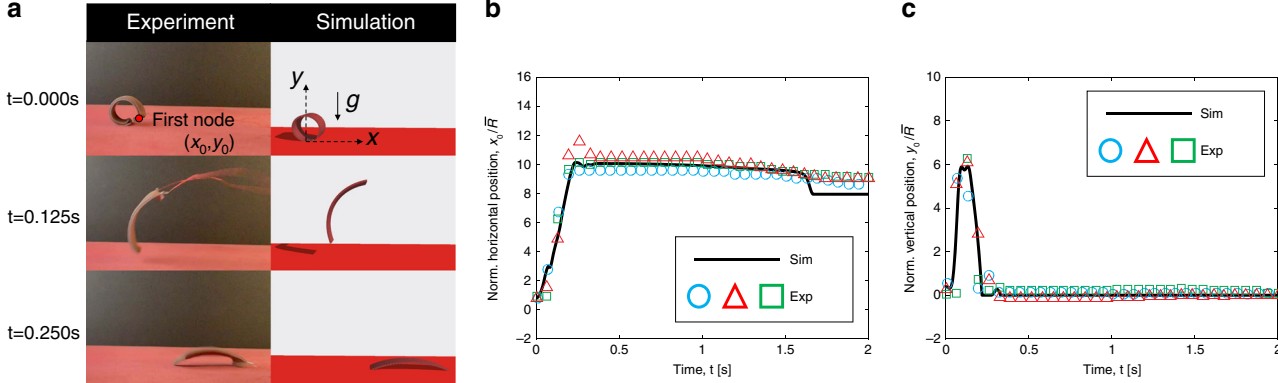

**Fig. 7 Dynamics of the jumper robot. a** Snapshots of the jumper robot from experiments and simulations at different time steps (Supplementary Movie 1). **b** Normalized $x$ position of the first node (marked in **a**) in jumper robot as a function of time from experiments (symbols) and simulations (solid lines). **c** Normalized $y$ position of the first node (marked in **a**) in jumper robot as a function of time from experiments (symbols) and simulations (solid lines).

friction, contact, and elastic/inelastic collision to achieve quantitative prediction of the motion of fast moving highly deformable soft robots. The computational efficiency makes it ideally suited for algorithms that iterate over a wide variety of parameters in order to select a robot design or locomotion strategy.

Overall, our results show good quantitative agreement between the simulations and experiments, suggesting that our numerical approach represents a promising step toward the ultimate goal of a computational framework for soft robotics engineering. However, further progress depends on additional experimental validation for a wider range of soft robot designs, locomotion gaits, and environmental conditions. The simulation introduced here also needs some prerequisite experimentally measured data, e.g., material properties of soft materials and their mechanical performance in response to external actuation. It would be meaningful to develop a more general constitutive relations that combines mechanics, electricity, heat, and magnetic field, for the direct simulation of soft robotic dynamics in response to external actuation. Moving forward, it would also be interesting to explore how DDG-based simulation tools that incorporate the formulation presented here can be used to generate optimal locomotion gaits that minimize cost of transport or maximize range for a prescribed energy input.

## Methods

**Fabrication of shape memory alloy actuators**. The fabrication process is similar to the one presented in refs. [42,46]. We start the fabrication process by laser-cutting two pieces of thermally conductive tape (H48-2, T-Global) with dimensions of $40 \times 18 \times 0.5$ mm ($55 \times 18 \times 0.5$ mm for the jumping robot) and $80 \times 55 \times 0.5$ mm by a $CO_2$ laser-cutting system (30 W VLS 3.50; Universal Laser Systems). Then we apply a layer of elastomer (Ecoflex 00-30, Smooth-On) that is prepared by mixing prepolymer at a 1:1 ratio by mass in a centrifugal mixer (AR-100, THINKY) with a thickness of 0.1 mm on top of the small thermally conductive tape and half-cure it in the oven under 50 °C for 7 minutes. Next, we place the pre-bent $\Omega$ shape SMA wire (0.015 inch diameter, $34 \times 11$ mm; Dynalloy) on top of the elastomer and apply another layer of elastomer with a thickness of 0.5 mm to encapsulate the SMA wire. Meanwhile, we stretch the larger thermally conductive tape ($80 \times 55 \times 0.5$ mm) to 150% of its original length and apply a 0.1 mm thick layer of elastomer on top of it. We place both thermally conductive tape in the oven under 50 °C for 7 minutes to half-cure them. After that, we attach the smaller thermally conductive to the middle of the stretched larger thermally conductive tape, clamp them with binder clips and place the bonded structure back to the oven for 10 minutes to fully cure it. Finally, we cut out the actuator along with the outline of the smaller thermally conductive tape.

**Experimental setup**. All robots and actuators in the experiment and characterization are powered by a desktop power supply (DIGI360, Electro Industries) under a current of 6 A. Each actuator is individually connected to an n-MOSFET (IRL7833, Nfineon for the rolling robot experiment and AO3416, Alpha & Omega Semiconductor for the rest) and the actuation and cooling time are controlled by a single-board microcontroller (Arduino UNO SMD R3, Arduino). The rolling robot and jumper experiment are performed on top of a red linatex sheet (MCM linatex 1.5 mm class 2, Linatex Corp of America).

## Data availability
The data that support the findings of this study are available from the corresponding author upon reasonable request.

## Code availability
Our numerical methods were implemented using the software available in the Supplementary Information of ref. [47].

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

## Acknowledgements

W.H. and M.K.J. acknowledge support from the Henry Samueli School of Engineering and Applied Science, University of California, Los Angeles. X.H. and C.M. acknowledge support from the Army Research Office (Grant #: W911NF-16-1-0148; Dr. Samuel Stanton) and the Office of Naval Research (ONR) under grant # N00014-17-1-2063 (Program Manager: Dr. Tom McKenna).

## Competing interests

The authors declare no competing interests.
