## [Peer Review File · Nature Communications]

Reviewers' Comments:

Reviewer #1:

Remarks to the Author:

The paper presents a discrete differential geometry based framework for simulation of soft robots made of shape memory alloys. Including elastic forces, damping forces, inertia, gravitational forces, collision contact and friction.

The modelling approach is based on "coarse" discretization (in comparison to most FEM frameworks) using simple rod/beam element formulations. The authors use an energy based formulation for deriving the elastic forces. The resulting equation of motion is solved using Newmark-beta time integration. Implementation of collision contact with a rigid surface is described as well as frictional contact (coulomb friction).

The modelling approach is tested on rolling/sliding ribbon. As study cases for validation a rolling robot and a jumping robot are used. Actuation is realized via shape memory alloys that change their natural curvature under the influence of Joule heating. To integrate actuation in the simulation the natural curvature of the elastic elements can be changed. Transient behaviour (heating and cooling) of the SMA is measured experimentally and approximated using piecewise fitted curves.

Results of simulations of the study cases rolling robot and jumping robot are validated qualitatively by comparing them visually to experimental results and quantitatively by evaluating the travelled distance of the robots.

The main contribution of the paper are on one side the relatively simple modelling approach that comprehensively captures the behaviour of the real system. On the other hand the low computation times offer great opportunity for application of the approach for design, optimization, control or motion planning.

The paper is well organized and written in a comprehensible way. Videos and figures additionally support the understanding of the work.

The supplementary material gives useful additional information regarding the computational time and derivation of parameters that are used in the model (stiffness, friction coefficient, transient actuation behaviour etc.)

It seems that experiments for the actuation behaviour of the shape memory alloys as well as for the rolling and jumping robot have only been conducted once each. For a stronger claim, please show the reliability of the robots' movements by repeated conduction of experiments.

Regarding the structure: Authors first describe the derivation of elastic forces then present the Newmark-beta integration for differential equations and then deal with the remaining forces. Why split the derivation of force terms?

Moreover, damping forces are referred to as external forces. Consider revising.

In Fig. 4 A there is a decay of Young's modulus and an increase of curvature even though the SMA is still being actuated/heated. Can you comment on that?

Fig 5 D2 shows that travelled distance after 4.5s is less than after 4s (simulation and experiment). In Fig 5 B this does not seem to be the case for simulation. The figures do not correspond.

All in all, the proposed framework with low computational cost provides a valuable contribution to the development of novel soft robots with slender structures and their application.

Reviewer #2:

Remarks to the Author:

This paper presents a simulation method adapted to a certain class of soft robots. The approach takes into account elasticity, dynamics, contact and friction with the environment.

The chosen mechanical modeling is based on discrete differential geometry techniques dedicated to slender structures.

The discretization of the problem is done by a set of nodes whose 2D positions constitute the degrees of freedom. So, contrary to what the images and the video can suggest, the method used is not 3D but allows to capture only a 2D problem.

In the same way the modeling of the friction is done in a cone of Coulomb only 2D... which greatly simplifies the problem.

At the level of the considered deformations, only two types are considered: stretching and bending. The associated energies are derived to deduce the associated forces and Hessian matrices.

Then an implicit (Newmark-beta) time integration is used to solve the problem. Damping forces are based on Rayleigh. 2D contact and friction forces are solved by writing the problem as a SOLCP and solved using the modified mass method. A status like method is used to solve the constraints.

Actuation is incorporated into the simulation by varying natural curvature and bending stiffness with time. A characterization of a single SMA actuator is described in the paper.

Different simulations are shown on the same robot with horizontal planar surface, inclined surface, uneven surface. But on each of these cases, the contact surface is not modeled as a surface but modeled by a line as the robot model is 2D. Finally a 2D simulation of a Jumper is done with another soft robot. For each of these cases, an experiment is performed, showing good correspondence with the simulation.

My general comment is that there is too few novelty in the paper (except the experiments and the comparison between the model and the soft robot). The state of the art is not sufficiently deeply studied to my opinion. In the state of the art in Computer Graphics, 3D deformable models and 3D collisions are largely diffused (see Bertails, Grinspun...) and methods to use these models in real-time (see Barbic).

In my team, we are developing a software in open source, that allows for 3D modeling of both deformation & friction contact of soft robots, in real-time.

<https://project.inria.fr/softrobot/examples-features/>

Corresponding papers:

- Coevoet, E., Morales-Bieze, T., Largilliere, F., Zhang, Z., Thieffry, M., Sanz-Lopez, M., ... & Duriez, C. (2017). Software toolkit for modeling, simulation, and control of soft robots. *Advanced Robotics*, 31(22), 1208-1224.

Another paper that demonstrates the use of a 2D model should also have cited and compared the method with:

Fei, Y., & Xu, H. (2016). Modeling and motion control of a soft robot. *IEEE Transactions on Industrial Electronics*, 64(2), 1737-1742.

Given the limitation of the presented approach (only 2D) and the lack of strong positioning with

existing work, I don't think that the paper should be published in such an excellent journal as Nature Communication.

But there are some interesting aspects in the results (the comparison between the model and the real robot is very good) and I sincerely advice the authors to re-submit the paper in a more accessible journal that allows for short papers (for instance Robotic and Automation Letters).

Reviewer #3:

Remarks to the Author:

The paper describes an approach to using discrete elastic rods (DER) to model a planar robot. It is an extension to ref. 33 that includes a more precise inertia and friction model. The authors describe their approach and force models and analyze the output of their simulation on 3 systems: a rolling ribbon, rolling robot, and jumping robot. The paper is clear, and the results look promising.

My main feedback would be for the authors to more precisely state what their contribution and the implications of their work are. The authors do a very nice job explaining the details of their numerical simulation. However, as they state, DER is commonly used in the computer graphics community for fast simulation. It is unclear from the text what the difference is between those methods and the one that the authors use.

Beyond the methods themselves, it seems to me that a major difference from computer graphics work is the direct comparison between the simulated results and experimental ones. The good agreement between the simulation and hardware experiments comes from a fit that the authors performed on the simulation parameters. The impact of the paper would be enhanced with a more general discussion on what parameters need to be experimentally verified and what the major limitations of the method would be if it were to be extended to a different robotic system.

== Other comments ==

- Starting the Numerical Simulation section with a discussion of the geometry of the rolling robot is strange, considering that the rest of the section is general. I would suggest moving it to the Rolling Robot section, and further also including a short description of the system geometry at the beginnings of the Rolling Ribbon and Jumper sections
- Given that the authors' main claim is to be able to model fast, dynamic soft robots, the focus on the rolling robot is strange. One gait cycle of the robot is 3s, and actuation takes up 0.25s of that time. How do the results differ from a model where, for example, we find the equilibrium state of the robot post-actuation and then simulate it forward as if it were a rigid asymmetric rimless wheel?
- On pg 16, "perpendicular" to gravity should be "parallel" or "antiparallel".
- It seems from the plots that the simulations consistently under-predict the motion of the wheel. Please explain.
- Considering one of the main claims is that the simulation can run faster than real-time, I would suggest moving some of the computation time experiments from the Supplementary Material into the article itself.
- What is the drop in x position that occurs in Fig. 6B around $t=1.5s$? It is not reflected in the experimental results and seems to occur after the robot has already returned to the ground.

- In the supplementary material, the authors state that the numerical method becomes unstable when $h > 10\text{ms}$. Could they include the data for this run in Fig. 7A (similarly to the $nv=49$ case in Fig. 7B)?

- The plots in the supplementary material, Fig. 7 are difficult to read. The quantitative analysis of the results would be much more convincing if the authors were able to report some summary statistics or metrics of this data.

Response to the Reviewers' comments for manuscript #:NCOMMS-19-539396
“Dynamic Simulation of Articulated Soft Robots ”
W. Huang, X. Huang, C. Majidi, and M. K. Jawed

Here, we provide a summary of the changes made to the revised manuscript, followed by a detailed response to each of the Reviewers' comments/suggestions (**in bold**). Please refer to the “diff. version” attached at the end of this document for a detailed account of all the changes, corrections, and additions to the revised versions of the manuscript. In that document, the original text that was modified/deleted is struck-through (in black) and the revised/new text is in blue.

1. Summary of changes made to the manuscript:

- To verify the robustness and accuracy of our desktop experiments, **we have conducted all experiments (rolling robot and jumper robot) multiple times** and included the data in the manuscript.
- We have rewritten the Introduction and Conclusion of the manuscript to emphasize the novelty of our results.
- We have added detailed discussion on the current limitations of our numerical method in the “Discussion” section.
- We now introduce all other force vectors, e.g. internal damping force, gravity, and reaction force, after the discussion of elastic force vector.
- We have modified “external damping force” as “internal damping force” for the description of viscoelastic behavior of soft material.
- We have explained the reason of the decay in the SMA response when discussing the mechanics of a single actuator.
- We have moved the geometric description of the soft rolling robot from the “Numerical Simulation” section to the “Rolling Robot” section, and described the geometric properties of the rolling ribbon and jumper at the beginning of each section.
- We have changed the image sequence of Fig. S2C and Fig. S2D in Supplementary Material (SM) in order to better explain our fitting process.
- We have modified the figure sequence in Fig. 5B to include $t=4.0s$ and $t=4.5s$.
- We have cited additional relevant literature in the “Introduction” section.
- We have moved the “computational time” section from SM to main article.
- We have included two tables in SM for better rendering the data.
- We fixed minor typographical errors and implemented stylistic corrections of our own.

2. Response to Reviewer I

Comment L1: The paper presents a discrete differential geometry based framework for simulation of soft robots made of shape memory alloys. Including elastic forces, damping forces, inertia,

gravitational forces, collision contact and friction. The modelling approach is based on "coarse" discretization (in comparison to most FEM frameworks) using simple rod/beam element formulations. The authors use an energy based formulation for deriving the elastic forces. The resulting equation of motion is solved using Newmark-beta time integration. Implementation of collision contact with a rigid surface is described as well as frictional contact (coulomb friction).

The modelling approach is tested on rolling/sliding ribbon. As study cases for validation a rolling robot and a jumping robot are used. Actuation is realized via shape memory alloys that change their natural curvature under the influence of Joule heating. To integrate actuation in the simulation the natural curvature of the elastic elements can be changed. Transient behavior (heating and cooling) of the SMA is measured experimentally and approximated using piecewise fitted curves.

Results of simulations of the study cases rolling robot and jumping robot are validated qualitatively by comparing them visually to experimental results and quantitatively by evaluating the travelled distance of the robots.

The main contributions of the paper are on one side the relatively simple modelling approach that comprehensively captures the behavior of the real system. On the other hand, the low computation times offer great opportunity for application of the approach for design, optimization, control or motion planning.

The paper is well organized and written in a comprehensible way. Videos and figures additionally support the understanding of the work.

The supplementary material gives useful additional information regarding the computational time and derivation of parameters that are used in the model (stiffness, friction coefficient, transient actuation behavior etc.)

It seems that experiments for the actuation behavior of the shape memory alloys as well as for the rolling and jumping robot have only been conducted once each. For a stronger claim, please show the reliability of the robots' movements by repeated conduction of experiments.

Response I.1: We thank the Reviewer for their comments and helpful feedback. Following their suggestions, we have conducted **all** the experiments multiple times during the revised version, and plot the simulation results and all experimental data in Fig. 5 D1-D6 (for rolling robot) and Fig. 7B and C (for jumper robot). For consistency, we also moved the plot of the rolling robot in the planar case (previously Fig. 4B) to Fig. 5 D1.

The modified Fig. 5 and Fig. 7 are shown in **Fig. R1** and **Fig. R2** individually.

Fig R1: Experimental data for rolling robot.

Fig R2: Experimental data for jumper robot.

In page 17, during the discussion of rolling robot in the planar case, we now note the following

“Note that the different symbols correspond to repeated experimental runs.”

Comment I.2: Regarding the structure: Authors first describe the derivation of elastic forces then present the Newmark-beta integration for differential equations and then deal with the remaining forces. Why split the derivation of force terms?

Response I.2: We thank for pointing it out. On the basis of the reviewer’s suggestion, we have switched the sequence of the derivations – i.e. we now discuss the damping forces, external gravity forces, and external reaction forces after the formulation of elastic forces. Also, we delete the explanation of these forces after the discussion of equations of motion.

On page 8, we have added the following subsection:

*“*Other force vectors*

Besides the internal elastic forces, F^s and F^b , the structure would also experience internal damping forces during deformation. We use the Rayleigh damping matrix to formulate the viscoelastic behavior of soft robots, such that the damping force vector is given by

$$F^d = -(\alpha M + \beta K)v$$

where $\alpha, \beta \in R^+$ are damping coefficients, $K = -\frac{\partial}{\partial q}(F^s + F^b)$ is the tangent stiffness matrix, and v is the velocity vector (time derivative of DOF). Also, the external gravity forces are denoted by F^g , as well as the external contact forces, F^r . The gradients of these force vectors can be analytically formulated in a manner similar to those of elastic forces.”

We also deleted the following after Eq.(2),

“The forces due to elastic stretching, F^s , and bending, F^b , can be evaluated from the energy gradient as discussed before; F^g is the external gravitational force; $F^d = -(\alpha M + \beta K)v$ is the damping force

(with $\alpha, \beta \in R^+$ and $K = -\frac{\partial}{\partial q}(F^s + F^b)$ is the tangent stiffness matrix); and F^r is the reaction force from the ground.”

Comment I.3: Moreover, damping forces are referred to as external forces. Consider revising.

Response I.3: The Reviewer is correct, we regret the mistake. We have now replaced the term “external damping force” with “internal damping forces” throughout the manuscript.

Comment I.4: In Fig. 4 A there is a decay of Young's modulus and an increase of curvature even though the SMA is still being actuated/heated. Can you comment on that?

Response I.4: The reviewer raises an important question. We fit the mechanical response of SMA (in Fig. 2D) by a piece-wise function on the basis of the results shown in Fig. S2C of the SM. Specifically, this fit contains a sharply changing linear component during the short time duration t_0 when SMA phase change occurs, and a smoothly changing function during the remainder of the actuation and cooling cycles. The primary reason for the slight increase in curvature and decrease in Young’s modulus after t_0 is that the fitting function is constrained to be smooth and monotonic (i.e. either increase, decrease, or remain constant). While we could separate the fitting into three or more piece-wise functions, this will lead to added complexity with little improvement in fitting accuracy.

In case of the curvature, the slight increase after time t_0 is also observed experimentally. This is particularly evident in Fig. S2C of the SM, where the actuator exhibits damped oscillation just after actuation. This arises because the martensite crystal phases within the SMA have almost entirely transformed to austenite during the short duration t_0 . Due to the actuator’s speed and inertia, it will typically overshoot and then exhibit a subsequent relaxation in curvature change.

To clarify these points in the manuscript, we have switched the sequence of Fig. S2C and Fig. S2D in SM in order to make it more clear that the piece-wise function was fitted using experimental data. Also, we now provide a more complete explanation for the decay modeled for the single actuator:

“Note that the parameter t_0 is not necessarily equal to the actuation time of 0.25s. As a result, the curvature slightly increases (and Young’s modulus decreases) even when the actuator is being heated (at $t_0 < t < 0.25$ s). The primary reason behind behavior is that the fitting function is constrained to be smooth and monotonic (i.e. either increase, decrease, or remain constant). While we could separate the fitting into more than two piece-wise functions involving more fitting parameters, this will lead to added complexity with little improvement in fitting accuracy.”

~~Also, as we fitted this piece-wise function based on experimental data, we switched the sequence of Fig. 2C and Fig. 2D in SM, to better explain the fitting process.~~

It should be noted that in our previous work, [Goldberg et al. *Soft Robotics* (2019).], an even simpler fitting function was used to characterize the acutated SMA in response to heat. This fit ignored the initial linear phase, as shown in Fig. 8a of that paper, leading to a less accurate prediction of the SMA response.

Comment I.5: Fig 5 D2 shows that travelled distance after 4.5s is less than after 4s (simulation and experiment). In Fig 5 B this does not seem to be the case for simulation. The figures do not correspond.

Response I.5: Thanks for pointing out this discrepancy. We had mistakenly ordered the sequence of figures in Fig. 5B from simulation side. We have now switched the sequence of the figure in the revised version.

Comment I.6: All in all, the proposed framework with low computational cost provides a valuable contribution to the development of novel soft robots with slender structures and their application.

Response I.6: We thank the reviewer for their valuable time and constructive feedback.

3. Response to Reviewer II

Comment II.1: This paper presents a simulation method adapted to a certain class of soft robots. The approach takes into account elasticity, dynamics, contact and friction with the environment.

The chosen mechanical modeling is based on discrete differential geometry techniques dedicated to slender structures.

The discretization of the problem is done by a set of nodes whose 2D positions constitute the degrees of freedom. So, contrary to what the images and the video can suggest, the method used is not 3D but allows to capture only a 2D problem.

In the same way the modeling of the friction is done in a cone of Coulomb only 2D... which greatly simplifies the problem.

At the level of the considered deformations, only two types are considered: stretching and bending. The associated energies are derived to deduce the associated forces and Hessian matrices.

Then an implicit (Newmark-beta) time integration is used to solve the problem. Damping forces are based on Rayleigh. 2D contact and friction forces are solved by writing the problem as a SOLCP and solved using the modified mass method. A status like method is used to solve the constraints.

Actuation is incorporated into the simulation by varying natural curvature and bending stiffness with time. A characterization of a single SMA actuator is described in the paper.

Different simulations are shown on the same robot with horizontal planar surface, inclined surface, uneven surface. But on each of this case, the contact surface is not modeled as a surface but modeled by a line as the robot model is 2D. Finally, a 2D simulation of a Jumper is done with

another soft robot. For each of these cases, an experiment is performed, showing good corresponding with the simulation.

My general comment is that there is too few novelty in the paper (except the experiments and the comparison between the model and the soft robot). The state of the art is not sufficiently deeply studied to my opinion. In the state of the art in Computer Graphics, 3D deformable models and 3D collisions are largely diffused (see Bertails, Grinspun...) and methods to use these models in real-time (see Barbic).

In my team, we are developing a software in open source, that allows for 3D modeling of both deformation & friction contact of soft robots, in real-time.

<https://project.inria.fr/softrobot/examples-features/>

Corresponding papers:

- Coevoet, E., Morales-Bieze, T., Largilliere, F., Zhang, Z., Thieffry, M., Sanz-Lopez, M., ... & Duriez, C. (2017). Software toolkit for modeling, simulation, and control of soft robots. *Advanced Robotics*, 31(22), 1208-1224.

Another paper that demonstrates the use of a 2D model should also have cited and compared the method with:

Fei, Y., & Xu, H. (2016). Modeling and motion control of a soft robot. *IEEE Transactions on Industrial Electronics*, 64(2), 1737-1742.

Given the limitation of the presented approach (only 2D) and the lack of strong positioning with existing work, I don't think that the paper should be published in such an excellent journal as *Nature Communication*.

But there are some interesting aspects in the results (the comparison between the model and the real robot is very good) and I sincerely advice the authors to re-submit the paper in a more accessible journal that allows for short papers (for instance *Robotic and Automation Letters*).

Response II.1: We thank the reviewer for sharing their concerns and feedback.

Our results show that if (i) Newmark-beta integration scheme (to prevent artificial energy loss), (ii) material damping (to capture inelastic collision with ground), (iii) Coulombic friction (to model frictional contact), and (iv) modified mass method (to incorporate unevenness of the substrate) are integrated with a discrete simulation framework, it can accurately capture the dynamics of a fast moving soft robot in real-time on a single thread of a contemporary desktop processor. The same framework can be applied in 3D rod-based models and even shell-based models (more on this later in the response).

In this early work on DDG-based robot simulation, we consider a 2D testbed and implement the appropriate numerical method. The two robotic testbeds - rolling robot and jumper - exhibit in-plane motion and no twisting deformation during the whole dynamic process.

In order to demonstrate the generality of the approach, we refer to Fig. R2 that presents a snapshot of a shell-based simulation of the rolling ribbon (Fig. 3 in the manuscript used a rod-based model) using discrete shell model (instead of a rod model). We hope to report soon the detailed numerical procedure for simulation of robots using shell model in a technical journal, e.g. *Robotics and Automation Letters* as suggested by the Reviewer. We restricted our results to the planar case in the manuscript under review primarily for brevity and clarity.

Figure R4. Rolling band simulation performed by Discrete Shell model with 3D contact.

While finite element-based simulation approach is essential for fundamental understanding of the actuation mechanism and locomotion modes, DDG-based simulations are powerful when dealing with system-level simulation of complex structures and contact/friction. This feature of DDG-based simulations, as well as its relative simplicity and computation time, has been reiterated in several seminal papers in the computer graphics community [Refs. 30-35 in the revised manuscript].

In order to emphasize the novelty of the simulation procedure, we have rewritten the “Introduction” section (second and third paragraphs on page 3-4). With regards to the novelties introduced in the current manuscript, we believe that this work is novel in the following respects (also see our response to comment III.1 of Reviewer III):

1. We used a modified mass-based method to account for the frictional contact and uneven surface. Usually, contact is modeled through an explicit approach, such that the simulator can only converge and give reliable predictions at smaller time step. Here, the Hessian matrix of the constraint system is also analytically given, such that we can use a larger time step to implicitly solve the Second Order Linear Complementarity Problem (SOLCP). As a result, the numerical framework is faster than real time.

2. Prior works on DER employed first order implicit Euler integration scheme, which suffers from artificial energy loss. While this is sufficient for visual realism, it is inadequate to capture the inertial effects - a key feature of fast moving soft robots. We incorporate a momentum preserving second order Newmark-beta integration scheme to avoid artificial energy dissipation. As evidenced by our experimental validation, this can successfully capture the inertia-dominated dynamics of soft robots.

3. We used a rate-dependent viscoelastic model to capture the elastic/inelastic collision between soft material and a rigid wall. Previous applications of the DER method model the slender structure as a linear elastic material [Bergou, Miklós, et al. "Discrete elastic rods." *ACM transactions on graphics (TOG)*. Vol. 27. No. 3. ACM, 2008.]. In doing so, such approaches do not give consideration to the rate-dependent viscoelastic property of soft material; as such, the collision and rebound between soft robot and rigid wall cannot be precisely described. Here, we employed Rayleigh damping matrix to model the rate-dependent viscoelastic behaviors of SMA. The rebound height is related to the material damping coefficient, simply because of the energy dissipation during the contact-compress-rebound process.

4. We perform precise experiments using soft robotic testbeds to quantitatively validate the correctness of our numerical framework. Simulations are performed based on constitutive properties from independent measurements of the mechanical response of the SMA during actuating-cooling process. The constitutive model is fed into the simulator and dynamic simulation of soft rolling robot and jumper robot is performed without the need for additional data fitting.

See Response III.1 for a summary of changes to the manuscript to focus the novelty of the numerical method.

We also added citations to the following relevant literature:

[Fei, Yanqiong, and Hongwei Xu. "Modeling and motion control of a soft robot." *IEEE Transactions on Industrial Electronics* 64.2 (2016): 1737-1742.]

[Coevoet, Eulalie, et al. "Software toolkit for modeling, simulation, and control of soft robots." *Advanced Robotics* 31.22 (2017): 1208-1224.]

[Grinspun, Eitan, et al. "Discrete differential geometry: an applied introduction." *ACM SIGGRAPH Course 7* (2006): 1-139.]

[Kaufman, Danny M., et al. "Adaptive nonlinearity for collisions in complex rod assemblies." *ACM Transactions on Graphics (TOG)* 33.4 (2014): 123.]

4. Response to Reviewer III

Comment III.1: The paper describes an approach to using discrete elastic rods (DER) to model a planar robot. It is an extension to ref. 33 that includes a more precise inertia and friction model. The authors describe their approach and force models and analyze the output of their simulation on 3 systems: a rolling ribbon, rolling robot, and jumping robot. The paper is clear, and the results look promising.

My main feedback would be for the authors to more precisely state what their contribution and the implications of their work are. The authors do a very nice job explaining the details of their numerical simulation. However, as they state, DER is commonly used in the computer graphics community for fast simulation. It is unclear from the text what the difference is between those methods and the one that the authors use.

Response III.1: Prior works in computer graphics on simulation of slender structures, e.g. rods, plates, shells, viscous threads, and viscous sheets, [Ref. 30-35 in the revised manuscript] exclusively focus on visual realism. None of these prior works provide quantitative comparison with experiments. In this work, our main contribution is the following: we identify the necessary physical ingredients (e.g. frictional contact, material damping, and inertial effects) required to simulate fast moving soft robots in a physically-accurate manner (as evidenced through detailed comparison with experiments). A numerical simulation procedure is then designed and implemented that incorporates these ingredients and still maintains *faster than real-time* computation speed.

Based on the Reviewer's feedback, we have substantially modified the "Introduction" (second and third paragraphs in the main text) and "Conclusion" sections; see the attached .diff version of the manuscript. Primary changes are reproduced below:

- We added the following sentences to para. 2 of "Introduction" (page 3): "*Despite the visual realism in these simulation methods, these prior works do not comprehensively capture all the physical ingredients for a physically accurate simulation of fast moving articulated soft robots. Our numerical method integrates these ingredients - frictional contact, material damping, and inertial effects - into a discrete simulation framework to achieve quantitative agreement with experiments.*"
- We rewrote a major portion of the third paragraph (page 4) to specifically point out the novel features of the simulation: "*Our approach employs a discrete representation of a soft robot and incorporates Coulomb frictional contact, inelastic collision with ground, and inertial effects in a physically accurate manner. The mechanical deformation of the robot is associated with local elastic (stretching and bending) energies at each discrete node. We formulate these discrete elastic energies and, subsequently, the discrete equations of motion representing the balance of*

forces using principles from classical elastic rod theories [29,38]. Coulomb frictional contact with uneven surface is integrated into the formulation using the modified mass method [33], such that a group of constrained equations of motion can be implicitly updated through a second order, symplectic Newmark-beta time integration scheme. Since this integration scheme is momentum preserving, it does not suffer from artificial energy loss - a well-known attribute of first order Euler integration used in prior works with discrete rod simulations [29] - and can capture the essential inertial effects during the dynamic simulation of soft robots. The elastic/inelastic collision between the soft robot and rigid ground can be captured by the rate-dependent viscoelastic behavior of the soft material, i.e., the damping coefficient in Rayleigh's damping matrix is used to precisely control the recovery factor during collision and rebound [39]. Finally, the experimentally measured data of a single actuator during one actuating-cooling cycle is fed into our numerical framework for the investigation of soft robotic dynamics. The result is a robust simulation tool that can potentially run faster than real-time on a single thread of a desktop processor. Convergence with spatial and temporal step size of the numerical method is quantitatively verified.”

- We rewrote the “Conclusion” section (page 22-23); the relevant portion reads: *“Specifically, our numerical framework was developed on the basis of a well-established Discrete Elastic Rods (DER) algorithm. Considering the geometric properties of our soft robotic structures, we simplified the traditional DER method in 2D and ignored the twisting curvature in slender structures, such that the computational efficiency can be significantly improved. To avoid the artificial energy dissipation during the time marching scheme, we replaced the first order, implicit Euler integration by a second order, symplectic Newmark-beta method, for the momentum preservation during the dynamic simulation. For the frictional contact between the rigid wall and the soft material, Coulomb’s law was implemented through a modified mass-based method, and this fully implicit framework allows a larger time step for convergence and numerical stability, which is also a prerequisite for real-time simulation. Similarly, the elastic/inelastic collision between the rigid wall and soft robots, related to the rate-dependent viscoelastic behavior of soft material, can be precisely described by Rayleigh damping matrix. The mechanical response of SMA during actuating-cooling process was first experimentally measured through a single actuator, then fed to the numerical framework to simulate the dynamics of the soft robots.”*

In addition to the changes above, we have substantially improved our experimental validation. As mentioned above in response to Review I, we have tripled the total number of experimental trials and included the expanded data set in the manuscript. The extensive validation of the numerical method should also help set us apart from the prior works in the computer graphics literature.

Comment III.2: Beyond the methods themselves, it seems to me that a major difference from computer graphics work is the direct comparison between the simulated results and experimental ones. The good agreement between the simulation and hardware experiments comes from a fit that the authors performed on the simulation parameters. The impact of the paper would be

enhanced with a more general discussion on what parameters need to be experimentally verified and what the major limitations of the method would be if it were to be extended to a different robotic system.

Response III.2: We appreciate the Reviewer's recommendation and have revised the manuscript to address these points. In general, the material parameters, e.g. Young's modulus, density, damping coefficients, and response to external actuations (heat, current, magnetic field), should be experimentally verified beforehand. In our case, we need to measure the mechanical response of SMA when actuated by current/heat.

The major limitation of our method is, we do not have a general, accurate constitutive relation that couples the mechanical response with other sources of external stimulation, e.g. temperature and magnetic field. In this manuscript, we explicitly measure the mechanical response of SMA when actuated by current, and then use the experimentally measured data as an input for our simulator. In the future, it would be interesting to develop a more general constitutive law that can couple the stress and heat/magnetic in slender structures.

Based on Reviewer's suggestion, we extensively modified the "Introduction" and "Conclusion" sections to address the novelty and limitation of our numerical framework,

In the "Introduction", we have added the following:

"The DDG approach starts with discretization of the smooth system into a mass-spring-type system, while preserving the key geometric properties of actual physical objects, and this type of simulation tool is naturally suited to account for contact and collision."

In "Conclusion" part, we emphasize on page 22,

"The simulation introduced here also needs some prerequisite experimentally measured data, e.g. material properties of soft materials and their mechanical performance in response to external actuation. It would be meaningful to develop a more general constitutive relation that combines mechanics, electricity, heat, and magnetic field, for the direct simulation of soft robotic dynamics in response to external actuation."

See our response to comment III.1 for a summary of additional changes.

== Other comments ==

Comment III.3: - Starting the Numerical Simulation section with a discussion of the geometry of the rolling robot is strange, considering that the rest of the section is general. I would suggest moving it to the Rolling Robot section, and further also including a short description of the system geometry at the beginnings of the Rolling Ribbon and Jumper sections

Response III.3: We changed the sequence based on reviewer's suggestion. We moved the subsection describing the geometry of the rolling robot to the *Rolling Robot* section. Also, we briefly discuss the geometric descriptions of the Rolling Ribbon and Jumper Robot at the beginning of each section.

In the "Rolling ribbon" section, we described the geometry used in our simulation,

"In the numerical study here, the arc length we chose for the circular ribbon is $L_0 = 0.3$ m, resulting in $\bar{R}_0 = 0.3/2\pi \approx 0.048$ m."

We moved the geometric details of the rolling robot to the "Rolling Robot" section:

"The star-shaped, rolling robot in Fig. 2A is comprised of seven compliant actuators/limbs that are arranged radially. Each limb has a curved part with length $l_c = 2.2$ cm and a straight part with length $l_s = 0.8$ cm. The natural curvature of the curved part is $\kappa_0 \equiv 1/R_c = 120\text{m}^{-1}$. The material density of the rolling robot is $\rho = 1912\text{kg/m}^3$. The mass center is located at (x_c, y_c) . The height, $H \approx 5$ cm, is used as the body length. We then discretize the structure into N nodes, shown schematically in Fig. 2A. This corresponds to a degrees of freedom (DOF) vector, $q = [x_0, y_0, \dots, x_{N-1}, y_{N-1}]^T$, of size $2N$, representing the vertical and horizontal coordinates of each node. Here, the superscript T denotes transposition. The length of each edge -- the segment between two consecutive nodes -- in this study is $\Delta l \approx 2.5$ mm, resulting in $N=84$ nodes (convergence study in the Supplementary Materials)."

In the "Jumper" section, we have added the following:

The SMA-based jumper shown in Fig. 7A is an asymmetric circle with radius $\bar{R}_0 \approx 0.05$ mm. Detailed geometric property of Jumper robot can be found in the Supplementary Materials.

Comment III.4: - Given that the authors' main claim is to be able to model fast, dynamic soft robots, the focus on the rolling robot is strange. One gait cycle of the robot is 3s, and actuation takes up 0.25s of that time. How do the results differ from a model where, for example, we find the equilibrium state of the robot post-actuation and then simulate it forward as if it were a rigid asymmetric rimless wheel?

Response III.4: We thank the reviewer for this interesting insight. As we experimentally measured, the actuating-cooling cycle for each actuator element is about 3s. Since the rolling robot has 7 actuators, we could greatly speed up motion by activating the actuators in $3/7 = 0.43$ s increments. However, we observe that this results in uncontrolled motion that does not reach steady-state and has poor repeatability. We selected the slower 3s gait cycle in order to ensure more reliable and repeatable steady-state measurements for validating the computational scheme.

The idea of modeling the rolling robot as a rigid rimless wheel is an intriguing one. A rigid body dynamic simulation would likely result in a visually realistic simulation. However, it will not match the experiments for the following reason. This would eliminate the crucial role of elasticity and viscoelasticity in the mechanics of the soft robotic system. This corresponds to zero material damping (β

parameter in Rayleigh damping matrix) and results in physically inaccurate motion. See Fig. S3 of the Supplementary Material for the effect of material damping.

Another way of simulating the “*rigid rimless wheel*” is to use a very high elastic stiffness in the numerical method presented in the manuscript. However, this would imply a *stiff simulation* and a smaller time step for convergence. This will decrease the computational efficiency of our numerical framework. It would be interesting to explore a rigid body dynamic simulation to test whether it can work for the rolling robot; however, our current approach appears to be adequate for capturing both nonlinear deformations and complex locomotions of the soft robotic system.

Comment III.5: - On pg 16, “perpendicular” to gravity should be “parallel” or “antiparallel”.

Response III.5: Thank you for pointing it out. We replace “perpendicular” with “parallel”:

“This is the simplest scenario presented in Fig.5 D1 (also Movie S1), where the surface normal is anti-parallel to gravity.”

Comment III.6: - It seems from the plots that the simulations consistently under-predict the motion of the wheel. Please explain.

Response III.6: That is a good point. We think the main reason is the thickness of robot limb. As shown in Fig. R5, the outer distance (green line) is almost 30% larger than the inner distance (while line), while in simulation side, the limbs are modeled as rod segments with zero thickness. That would be the main reason for the under-prediction of the rolling robot motion.

At the end of the “rolling robot” section, we now include this reason for the under-prediction:

“However, we should also note that our simulator always under-predicts the motion of the rolling robot. We attribute this to the finite thickness of the actuator elements, which is not accounted for in the model.”

Fig. R5: geometry of soft rolling robot.

Comment III.7: - Considering one of the main claims is that the simulation can run faster than real-time, I would suggest moving some of the computation time experiments from the Supplementary Material into the article itself.

Response III.7: In the revised manuscript, we moved the “computational time” section from the SM to the main article.

After the discussion of “Rolling robot”, we add a new section to address the “computational time”,

“Our novel numerical tool can achieve real-time simulation of the soft rolling robot. In Fig. 6, with a fixed number of vertices, $N=84$, the computational time linearly scales with time step size h for all the scenarios. The simulations ran on a single thread of AMD Ryzen 1950X CPU @ 3.4 GHz. Also, our simulator can run faster than real time when the time step size $h \geq 2.5$ ms. Numerical issues associated with a large step size appear at $h \geq 10$ ms, in which case the computation time is infinite because we cannot get convergence. In summary, the numerical framework can quantitatively predict the motion of soft robots while running faster than real time on one thread of a contemporary desktop processor.”

Comment III.8: - What is the drop in x position that occurs in Fig. 6B around $t=1.5s$? It is not reflected in the experimental results and seems to occur after the robot has already returned to the ground.

Response III.8: We thank the Reviewer for their careful attention to this result. Even though the robot is already on the ground at $t=1.5s$, the SMA is not totally cooled (for reference, see Fig. S2C in the SM). Therefore, the actuator will continue to relax to its natural shape, resulting movement on the ground and change in the x-pos. Whereas, the change of x-pos in experiments is smooth, the change is sudden in the simulation. We attribute this mismatch between experiment and simulation to the limitations of using Coulomb’s law.

At the end of the jumper section, we explain the mismatch between the simulation and experiment,

“We should also note that, at time $t=1.5s$, the x position predicted by numerical simulation has a sudden drop, which is not reflected in the experimental data. The mismatch between numerical simulation and experimental observation is due to the discontinuity in Coulomb’s law, i.e. the jumper robot would move only if the external reaction force is out of the frictional cone.”

Comment III.9: - In the supplementary material, the authors state that the numerical method becomes unstable when $h>10ms$. Could they include the data for this run in Fig. 7A (similarly to the $nv=49$ case in Fig. 7B)?

Response III.9: The simulation cannot work at $h>10ms$, so we cannot get any numerical data for this case.

We added the following to the “Computation time” section:

“... in which case the computation time is infinite because we cannot get convergence.”

Comment III.10: - The plots in the supplementary material, Fig. 7 are difficult to read. The quantitative analysis of the results would be much more convincing if the authors were able to report some summary statistics or metrics of this data.

Response III.10: We transposed the Fig. S6 in SM for better rendering of the data. We have also included two tables in SM as an alternative way of reporting the data (see also Fig. R6).

Table 1: Final normalized centroid x coordinate, x_c/H .

time step size h	planar, $\theta = 0.0^\circ$	inclined, $\theta = +3.0^\circ$	Sin curve, crest	Sin curve, trough
$h = 5\text{ms}$	1.1660	1.0958	2.0012	0.3206
$h = 1\text{ms}$	1.1347	1.0975	1.9549	0.3168
$h = 0.5\text{ms}$	1.1356	1.1037	1.9577	0.3122
$h = 0.1\text{ms}$	1.1327	1.1050	1.9688	0.3025
$h = 0.05\text{ms}$	1.1373	1.0993	1.9672	0.3096

Table 2: Final normalized centroid x coordinate, x_c/H .

Number of vertices	planar, $\theta = 0.0^\circ$	inclined, $\theta = +3.0^\circ$	Sin curve, crest	Sin curve, trough
$nv = 49$	1.1055	1.0720	1.9289	0.0444
$nv = 84$	1.1327	1.1050	1.9688	0.3025
$nv = 98$	1.1356	1.0810	1.9598	0.3033
$nv = 105$	1.1528	1.0872	1.9740	0.2945
$nv = 112$	1.1573	1.1085	1.9738	0.3035

Fig. R6: Tables of convergence study.

Dynamic Simulation of Articulated Soft Robots

Weicheng Huang^{1†}, Xiaonan Huang^{2†}, Carmel Majidi^{2*}, and M. Khalid Jawed^{1*}

¹*Department of Mechanical and Aerospace Engineering, University of California, Los Angeles, 420 Westwood Plaza, Los Angeles, CA 90095.*

²*Department of Mechanical Engineering, Carnegie Mellon University, 5000 Forbes Avenue, Pittsburgh, PA 15213.*

[†] *W.H. and X.H. contributed equally to this work.*

**To whom correspondence should be addressed:*

cmajidi@andrew.cmu.edu and khalidjm@seas.ucla.edu.

Soft limbed robots are primarily composed of soft and deformable materials that can allow for mechanically robust maneuvers that are not typically possible with conventional, piece-wise rigid robotic systems. However, owing to the current limitations in simulation and modeling, design and control of soft robots often involve a painstaking trial and error process. With the ultimate goal of a computational framework for soft robotic engineering, we introduce a numerical simulation tool for limbed soft robots that draws inspiration from discrete differential geometry based simulation of slender structures widely used in the computer graphics community. In this framework, the limbs of the soft robot are treated as elastic rods that undergo continuous flexural deformation in response to actuator stimulation and surface tractions. The simulation incorporates an implicit treatment of the elasticity of the limbs, inelastic collision between a soft body and rigid surface, and unilateral contact

and Coulombic friction with an uneven surface. The computational efficiency of the numerical method enables it to run faster than real-time on a single thread of a desktop processor. For experimental validation, we first characterize the mechanical response of a single Shape Memory Alloy (SMA)-based actuator, and then design soft robots comprised of compliant limbs that can change shape upon thermal actuation through electrical Joule heating. Our experiments and simulations show quantitative agreement and indicate the potential role of predictive simulations for soft robot design and gait selection.

Introduction

Robots composed of soft and elastically deformable materials can be engineered to squeeze through confined spaces ¹, sustain large impacts ², execute rapid and dramatic shape change ³, and exhibit other robust mechanical properties that are often difficult to achieve with more conventional, piece-wise rigid robots ⁴⁻¹⁰. These platforms not only exhibit unique and versatile mobility for applications in biologically-inspired field robotics, but can also serve as a testbed for understanding the locomotion of soft biological organisms. In particular, they can help lead to new insights about the interplay between dynamics, actuator/muscle activation, mechanical deformation, and tribology that arise during locomotion of soft bodies within an unstructured environment. However, due to the current limitations with simulating the dynamics of soft material systems, design and control of soft robots often involve a painstaking trial and error process, and it can be difficult to relate qualitative observations to underlying principles of kinematics, mechanics, and tribology. Progress, therefore, depends on a computational framework for deterministic soft robot modeling

that can aid in design, control, and experimental analysis.

Previous efforts to simulate soft robots have focused on Finite Element Method ^{11,12}, reduced-order Finite Element Method 13-16¹³⁻¹⁶, voxel-based discretization ^{17,18}, and modeling of slender soft robot appendages using Cosserat rod theory 19-21¹⁹⁻²¹. Here, drawing inspiration from simulation techniques based on Discrete Differential Geometry (DDG) that are widely used in the computer graphics community ²², we introduce a DDG-based numerical simulation tool for examining the locomotion of limbed soft robots. The DDG approach starts with discretization of the smooth system into a mass-spring-type system, while preserving the key geometric properties of actual physical objects, and this type of simulation tool is naturally suited to account for contact and collision ²³. In particular, we treat the robot as being composed of multiple slender actuators that can be modeled using elastic rod theories as had been previously shown for soft and compliant actuators ²⁴⁻²⁸. In order to achieve rapid simulation runtimes, we adapt fast and efficient physically-based computational techniques that have gained traction within the computer graphics community to model slender structures, e.g. rods ²⁹⁻³¹, ribbons ³², plates ³³, shells ³⁴, viscous threads ^{30,35}, and viscous sheets ³⁶. Despite the visual realism in these simulation methods, these prior works do not comprehensively capture all the physical ingredients for a physically accurate simulation of fast moving articulated soft robots. Our numerical method integrates these ingredients – frictional contact, material damping, and inertial effects – into a discrete simulation framework to achieve quantitative agreement with experiments. Recently, a DDG-based formulation was used to model a caterpillar-inspired soft robot in which the individual segments of the robot were treated as curved elastic rod elements ³⁷. Although promising, this formulation could not accurately capture inertial

effects – a key feature of fast moving robots – and did not incorporate the necessary contact and friction laws required to achieve quantitative agreement with experimental measurements.

Our approach employs a discrete representation of a soft robot ~~in which mechanical deformation~~ and incorporates Coulomb frictional contact, inelastic collision with ground, and inertial effects in a physically accurate manner. The mechanical deformation of the robot is associated with local elastic (stretching and bending) energies at each discrete node. We formulate these discrete elastic energies and, subsequently, the discrete equations of motion representing the balance of forces using principles from classical elastic rod theories^{29,38}. Coulomb frictional contact with uneven surface is integrated into the formulation using the modified mass method³³. ~~such that a group of constrained equations of motion can be implicitly updated through a second order, symplectic Newmark-beta time integration scheme. Since this integration scheme is momentum preserving, it does not suffer from artificial energy loss – a well-known attribute of first order Euler integration used in prior works with discrete rod simulations²⁹ – and can capture the essential inertial effects during the dynamic simulation of soft robots.~~ The elastic/inelastic collision between ~~soft material the soft robot~~ and rigid ground can be captured by ~~Rayleigh’s classical damping; we use the material~~ the rate-dependent viscoelastic behavior of the soft material, i.e., the damping coefficient in ~~the Rayleighdamping matrix to~~ Rayleigh’s damping matrix is used to precisely control the recovery factor during collision and rebound³⁹. Finally, the experimentally measured data of a single actuator during one actuating-cooling cycle is fed into our numerical framework for the investigation of soft robotic dynamics. The result is a robust simulation tool that can potentially run ~~in faster than~~ real-time on a single thread of a desktop processor. Convergence with spatial and

Figure 1: Rolling robot: snapshots from simulations (left) and experiments (right) between (a) $t = 0$ to (e) $t = 2.00$ sec. Limb #4 is actuated for a rolling motion. The width (out-of-plane dimension in this figure) is 18mm.

temporal step size of the numerical method is quantitatively verified.

The reliability of this simulation tool for making quantitative predictions is systematically examined using three test cases. First, we demonstrate that three empirically-observed motion patterns of a deformable *rolling ribbon*⁴⁰ on a declined surface can be captured by our simulator. Next, we build two types of soft robots made of SMA-based limb: a star-shaped *rolling robot* composed of seven radially oriented limbs (Fig. 1) and a *jumper* robot with a single limb. The SMA-based robots were selected because of the ability to achieve rapid dynamic motions in which

both material deformation and inertia have a governing role^{41,42}. In order to examine the influence of friction and ground topology, locomotion experiments were performed on flat, inclined/declined, and wavy/undulating surfaces. In all cases, we found reasonable quantitative agreement between experimental measurements of the robot displacement and predictions obtained from the numerical simulation.

Figure 2: (A) Geometric discretization of soft rolling robot. (B) The bending curvature at i -th node is associated with an osculating circle of radius R_i that is constructed by projecting perpendicular lines from the edges. From elementary geometry, the curvature is $\kappa_i = 1/R_i = 2 \tan(\phi_i/2)/\Delta l$ ³⁰. (C) Coulomb law for frictional contact⁴³.

Numerical simulation

* Geometry of rolling robot

The star-shaped, rolling robot in Fig. 2A is comprised of seven compliant actuators/limbs that are arranged radially. Each limb has a curved part with length $l_c = 2.2\text{cm}$ and a straight part with length $l_s = 0.8\text{cm}$. The natural curvature of the curved part is $\bar{\kappa}_0 \equiv 1/R_c = 120\text{m}^{-1}$. In this section, we review the numerical framework that incorporates elasticity, contact with uneven surface, friction, and inelastic collision for a comprehensive soft robot simulator. The material density of rolling robot is $\rho = 1912\text{ kg/m}^3$. The mass center is located at (x_c, y_c) . The height, $H \approx 5\text{ cm}$, is used as the body length.

Our numerical method discretizes the structure into N nodes, shown schematically in Fig. 2A. This corresponds to a degrees of freedom (DOF) vector, $\mathbf{q} = [x_0, y_0, \dots, x_{N-1}, y_{N-1}]^T$, of size $2N$, representing the vertical and horizontal coordinates of each node. Hereafter, the superscript T denotes transposition. The length of each edge — the segment between two consecutive nodes — in this study is $\Delta l \approx 2.5\text{ mm}$, resulting in $N = 84$ nodes (convergence study in the Supplementary Materials). In this discrete setting, the robot is represented by a lumped mass at each node and associated elastic stretching and bending energies — reminiscent of a mass-spring system. Since the motion of the robot remains in 2D, we do not include a twisting energy of the rod, although this can be readily integrated into our framework²⁹. In this section, we review the numerical framework that incorporates elasticity, actuation, contact with uneven surface, friction, and inelastic collision for a comprehensive soft robot simulator. Starting from the discrete representation of elastic en-

ergies, we formulate equations of motion at each node and update the configuration of the **robot structure** (i.e. position of the nodes) in time by a second order, implicit Newmark-beta approach.

* Discrete energies and elastic forces

The rod segment between two consecutive nodes is an *edge* that can stretch as the robot deforms – analogous to a linear spring. The turning angle ϕ_i (see Fig. 2B) at node \mathbf{x}_i between two consecutive edges can change – similar to a torsional spring. The elastic energy from the strains in the robot can be represented by the linear sum of two components: (1) stretching energy of each edge and (2) bending energy associated with variation in the turning angle at the nodes. The discrete stretching energy at the edge connecting \mathbf{x}_i and \mathbf{x}_{i+1} is $E_i^s = \frac{1}{2}EA\varepsilon_i^2\Delta l$, where EA is the stretching stiffness (calculated as the product of the material elastic modulus E and actuator cross-sectional area A) and $\varepsilon_i = |\mathbf{x}_{i+1} - \mathbf{x}_i|/\Delta l - 1$ is the axial stretch. Associated with each turning angle ϕ_i is the discrete bending energy $E_i^b = \frac{1}{2}EI(\kappa_i - \bar{\kappa}_i)^2\Delta l$, where EI is the bending stiffness, $\kappa_i = 2\tan(\phi_i/2)/\Delta l$ is the curvature [Fig. 2B], and $\bar{\kappa}_i$ is the natural curvature (i.e. curvature evaluated in undeformed configuration). In the special case of a joint node where three edges meet, the bending energy is comprised of two components: one corresponding to the turning angle between the first and second edges and the second one arises from the turning angle between the second and third edges. The total stretching energy of the robot can be obtained simply by summing over all the edges, i.e. $E^s = \sum_i E_i^s$, and, similarly, the total bending energy is $E^b = \sum_i E_i^b$. In both experiments and simulations, we observe that the structure is nearly inextensible and the prominent mode of deformation is bending. We evaluated the bending stiffness by quantifying the

shape of an actuator under vertical load, as shown in the Supplementary Materials.

The elastic stretching (and bending) forces acting on a node \mathbf{x}_i can be obtained from the gradient of the energies, i.e. $-\left[\frac{\partial E^s}{\partial x_i}, \frac{\partial E^s}{\partial y_i}\right]^T$ (and $-\left[\frac{\partial E^b}{\partial x_i}, \frac{\partial E^b}{\partial y_i}\right]^T$). An implicit treatment of the elastic forces requires calculation of the $2N \times 2N$ Hessian matrix of the elastic energies. Other than the seven *joint* nodes that are connected with three other nodes, a node \mathbf{x}_i is only coupled with the adjacent nodes \mathbf{x}_{i-1} and \mathbf{x}_{i+1} in the discrete energy formulation. This results in a banded Hessian matrix with 6×6 blocks of non-zero entries along the diagonal. The only off-diagonal non-zero entries correspond to the seven joint nodes. The analytical expressions for the gradient and Hessian of the elastic energies can be found in Refs. ^{29,30}.

* Other force vectors

Besides the internal elastic forces, \mathbf{F}^s and \mathbf{F}^b , the structure would also experience internal damping forces during deformation. The We use the Rayleigh damping matrix to formulate the viscoelastic behavior of soft robots, such that the damping force vector is given by ³⁹

$$\mathbf{F}^d = -(\alpha\mathbb{M} + \beta\mathbb{K})\mathbf{v}, \quad (1)$$

where $\alpha, \beta \in \mathbb{R}^+$ are damping coefficients, $\mathbb{K} = -\frac{\partial}{\partial \mathbf{q}}(\mathbf{F}^s + \mathbf{F}^b)$ is the tangent stiffness matrix, and \mathbf{v} is the velocity vector (time derivative of DOF). Also, the external gravity forces are denoted by \mathbf{F}^g , as well as the external contact forces, \mathbf{F}^f . The gradients of these force vectors can be analytically formulated in a manner similar to those of the elastic forces. The sparse nature of the Hessian-Jacobian matrix is critical for computational efficiency during the solution of the equations

of motion, described next.

* Equations of motion

The DOF vector can be updated from current time step (t_k) to the next ($t_{k+1} = t_k + h$), $\mathbf{q}^{k+1} = \mathbf{q}^k + \Delta\mathbf{q}^{k+1}$, by a second-order, implicit Newmark-beta time integration ³⁹,

$$\begin{aligned}\Delta\mathbf{q}^{k+1} - h\mathbf{v}^k &= \frac{h^2}{4}\mathbb{M}^{-1}(\mathbf{F}^{k+1} + \mathbf{F}^k) \\ \Delta\mathbf{q}^{k+1} &= \frac{h}{2}(\mathbf{v}^{k+1} + \mathbf{v}^k) \\ \Delta\mathbf{v}^{k+1} &= \mathbf{v}^{k+1} - \mathbf{v}^k,\end{aligned}\tag{2}$$

where the velocity vector (time derivative of DOF) is \mathbf{v} , superscript $k+1$ (and k) denotes evaluation of the quantity at time t_{k+1} (and t_k), \mathbb{M} is the diagonal mass matrix, h is the time step size, and $\mathbf{F} = (\mathbf{F}^s + \mathbf{F}^b + \mathbf{F}^g + \mathbf{F}^d + \mathbf{F}^r)$ is the sum of elastic, damping, and external forces defined ~~next~~. ~~The forces due to elastic stretching, \mathbf{F}^s , and bending, \mathbf{F}^b , can be evaluated from the energy gradient as discussed before; \mathbf{F}^g is the external gravitational force; $\mathbf{F}^d = -(\alpha\mathbb{M} + \beta\mathbb{K})\mathbf{v}$ is the damping force (with $\alpha, \beta \in \mathbb{R}^+$ and $\mathbb{K} = -\frac{\partial}{\partial\mathbf{q}}(\mathbf{F}^s + \mathbf{F}^b)$ is the tangent stiffness matrix); and \mathbf{F}^r is the reaction force from the ground~~before. In the absence of dissipative forces and external contact forces, this method is symplectic and momentum preserving ^{39,44,45} – a critical feature for simulation of robots where inertial effects are significant.

* Contact and friction

Since soft robots are often intended for locomotion on unstructured terrain, we require a method to account for contact and friction with the ground. Importantly, the surface normal can

vary with the horizontal x -axis. We model the nonpenetration constraints and frictional contact forces that resist sliding along interfaces based on Coulomb's law. At each time step, we apply continuous collision detection to the predicted trajectory to gather contact constraints into a contact set \mathbb{C} , shown in Fig. 2C. For these calculations, the velocity $\mathbf{u} = [\mathbf{v}_{2j-1}, \mathbf{v}_{2j}]^T$ (subscript denotes element number in a vector), and the reaction force $\mathbf{R} = [\mathbf{F}_{2j-1}^r, \mathbf{F}_{2j}^r]^T$, at the j -th node (the contact point) satisfy the condition

$$\mathbb{C}(\mathbf{u}, \mathbf{R}) \Leftrightarrow \begin{cases} \mathbf{R} = \mathbf{0} \text{ and } u^\perp > 0 \text{ (taking off)} \\ R^\parallel < \mu R^\perp \text{ and } \mathbf{u} = \mathbf{0} \text{ (sticking)} \\ R^\parallel = \mu R^\perp \text{ and } u^\perp = 0 \text{ (sliding)}, \end{cases} \quad (3)$$

where $\mu = 0.8$ is the friction coefficient characterized by experiments (Supplementary Materials), and the superscript \parallel (and \perp) denotes the component along (and perpendicular to) the ground. At the normal and tangential subspaces of a contact node \mathbf{x}_j , we either know its perpendicular velocity u^\perp (u^\parallel for tangential component) or the perpendicular reaction force R^\perp (R^\parallel for tangential component), so the Coulombic frictional contact law can be treated as a Second Order Linear Complementary Problem (SOLCP) ⁴³. We employ the modified mass method ³³ to solve this SOLCP such that a contact node \mathbf{x}_j can be free (degrees of freedom is 2, *taking off*), constrained along the normal to the ground \mathbf{p} (degrees of freedom is 1, *sliding*), or fully constrained (degrees of freedom is 0, *sticking*). The two modified equations of motion for the j -th node ($j = 1, \dots, N$)

are

$$\begin{bmatrix} \mathbb{F}_{2j-1} \\ \mathbb{F}_{2j} \end{bmatrix} \equiv \begin{bmatrix} \Delta \mathbf{v}_{2j-1}^{k+1} \\ \Delta \mathbf{v}_{2j}^{k+1} \end{bmatrix} - \frac{h}{2M_j} \mathbf{S}^{k+1} \left(\begin{bmatrix} \mathbf{F}_{2j-1}^{k+1} \\ \mathbf{F}_{2j}^{k+1} \end{bmatrix} + \begin{bmatrix} \mathbf{F}_{2j-1}^k \\ \mathbf{F}_{2j}^k \end{bmatrix} \right) - \Delta \mathbf{z}^{k+1} = \mathbf{0}, \quad (4)$$

where \mathbb{F}_{2j-1} is the left hand side of the $(2j - 1)$ -th equation of motion, M_j is the mass associated with j -th node, $\Delta \mathbf{z}^{k+1}$ is the change in velocity we want to enforce along the constrained direction(s), and the modified mass matrix is

$$\mathbf{S}^{k+1} = \begin{cases} \mathbb{I} & \text{if ndof} = 2, \\ (\mathbb{I} - \mathbf{p}\mathbf{p}^T) & \text{if ndof} = 1, \\ \mathbf{0} & \text{if ndof} = 0, \end{cases} \quad (5)$$

where ndof is the number of free DOF at j -th node and \mathbb{I} is the 2×2 identity matrix. Note that when a node is free, $\Delta \mathbf{z}^{k+1} = \mathbf{0}$, and Eq. 4 reduces to Eq. 2. If the node is fully constrained ($\mathbf{S}^{k+1} = \mathbf{0}$), Eq. 4 reduces to $\Delta \mathbf{v}_j^{k+1} = \Delta \mathbf{z}^{k+1}$ and the change in velocity (as well as the position) is enforced to take the value prescribed by $\Delta \mathbf{z}^{k+1}$.

The solution to the $2N$ equations of motion in Eq. 4 starts with an initial guess $(\Delta \mathbf{v}^{k+1})^{(0)}$ and subsequent Newton's iterations to improve the solution until a desired tolerance is achieved:

$$(\Delta \mathbf{v}^{k+1})^{(n+1)} = (\Delta \mathbf{v}^{k+1})^{(n)} - \mathbb{J}^{(n)} \setminus \mathbb{F}^{(n)}, \quad (6)$$

where $\mathbb{J}^{(n)} = \frac{\partial \mathbb{F}}{\partial (\Delta \mathbf{v}^{k+1})}$ is the Jacobian matrix evaluated at $(\Delta \mathbf{v}^{k+1})^{(n)}$. The non-trivial terms in the evaluation of this Jacobian are the Hessian matrices of the elastic energies; the analytical expressions are available in Refs. ^{29,30}. Due to the presence of the ground, we need to check whether the new solutions, e.g. \mathbf{q}^{k+1} , \mathbf{v}^{k+1} and $(\mathbf{F}^r)^{k+1}$ (computed from force balance), satisfy the following conditions:

- A node \mathbf{x}_j cannot fall below the ground.

- The normal component of reaction force R^\perp exerted by the ground on a node \mathbf{x}_j must be along the outward normal to the surface, e.g. $R^\perp > 0$.
- The reaction force \mathbf{R} should be in the frictional cone zone K_μ (see Fig. 2C); if the reaction force is on the boundary of the cone, this node is allowed to slide along the tangential direction of surface opposite to reaction force, $\mathbf{u} \cdot \mathbf{R} < 0$.
- If the tangential velocity u^\parallel at a sliding node \mathbf{x}_j changes its direction, $(u^\parallel)^k \cdot (u^\parallel)^{k+1} < 0$, this node should be fully constrained.

If one of the above rules is broken, we rewind the simulation, add (or delete) constraints at the contact pair, and re-solve Eq. 4 with a new guess.

* Inelastic collision

When an elastic body drops onto a rigid surface, the motion normal to the surface of the contact nodes are constrained, the normal velocities are set to zero, and the tangential velocities are reduced based on impulse theory, $\Delta u^\parallel = \mu \Delta u^\perp$. If the structure is modeled as an ideal mass-spring system without viscoelasticity, the whole structure will rebound to a certain height and the recovery factor – the ratio of rebound to initial height – is not deterministic. This arises because the structure’s kinetic energy will transfer into elastic potential energy during compression and then convert back to kinetic energy during the rebound phase³⁹. We must account for the rate-dependent viscoelasticity of contact for predictive simulation, where the energy loss of the collision-compression-rebound process results in a deterministic rebound height. In the Sup-

plementary Materials, we show that the decrease in rebound height of the rolling robot can be determined by the parameter β in damping force $\mathbf{F}^d = -(\alpha\mathbb{M} + \beta\mathbb{K})\mathbf{v}$, such that the recovery factor of collision is also related to β . Physically, β represents a damping that opposes elastic deformation, without penalizing rigid body motion. Opposition to rigid body motion and momentum dissipation can be accounted by the viscosity α .

The overall numerical framework thus accounts for inertia, friction, and collision and shows good convergence with both time and space discretization, as outlined in the Supplementary Materials. This simulation can potentially run in real-time on a single thread of a contemporary desktop processor; details on computational time can be found in the Supplementary Materials is discussed later.

Rolling ribbon

Before examining soft robot locomotion, we first investigate the simpler motion of a circular ribbon on a declined surface in order to test the accuracy of numerical implementation of friction and contact. In the numerical study here, the arc length we chose for the circular ribbon is \$L_0 = 0.3\text{m}\$, resulting in \$R = 0.3/2\pi \approx 0.048\text{m}\$. Because of gravity, this close-loop elastic structure will first undergo transient dynamics and then, as shown in Fig. 3A, move with a steady state configuration. See the Supplementary Materials for the detailed physical parameters of the *rolling ribbon*. The final shape is determined by the ratio $\Gamma_g = L_g/R$ between the gravito-bending length scale $L_g = (EI/\rho g A)^{1/3}$ and the ribbon undeformed radius R ⁴⁰. In Fig. 3B (Inset), we plot the static

configurations of rolling ribbon at different values of Γ_g . At small values of Γ_g , the ribbon shows relatively large deformation with large region of contact. As Γ_g increases, the deformed shape becomes closer to its original undeformed shape and the contact length decreases to reach a single point at $\Gamma_g = \infty$.

Now we turn to the motion of a rolling ribbon. Three different motion patterns exist on a declined surface: pure sliding (Fig. 3A1), combined sliding and rotation (Fig. 3A2), and pure rotation (Fig. 3A3), depending on a dimensionless number, $\mu/\tan\theta$, where μ is the frictional coefficient and θ is the decline angle. In Fig. 3B, we show the ratio between the distance traveled by a point on the ribbon (red mark in Figs. 3A) and the ribbon centroid, δ , as a function of normalized

Figure 3: Motion patterns of elastic ribbons: (A1) pure sliding ($\mu/\tan\theta = 0$); (A2) combination of sliding and rotating ($0 < \mu/\tan\theta < 1$); and (A3) pure rotating ($\mu/\tan\theta \geq 1$). The configurations are measured at time $t = 10.0s$ and $t = 10.1s$. (B) The ratio between the route of ribbon boundary point and ribbon centroid, $\delta = s_b/s_c$, as a function of normalized frictional coefficient $\mu/\tan\theta$, for different values of Γ_g .

friction coefficient, $\mu/\tan\theta$, at different values of Γ_g . When the normalized frictional coefficient $\mu/\tan\theta = 0$, the ribbon will slide along the tangential direction of the surface without any rotation, and the path of boundary point is the same as the path of center, $\delta = 1$. If $0 < \mu/\tan\theta < 1$, the motion of the ribbon is a combination of sliding and rotation, and the larger the friction, the higher the δ . The ribbon undergoes pure rotation at $\mu/\tan\theta \geq 1$ when δ remains fixed at a constant value depending on Γ_g . At the limiting case of a rigid ribbon, the motion is purely rotational and any point on the ribbon traces a cycloid path, corresponding to $\delta = 8/2\pi$. We plot the boundary node position as a function of time for all three cases in the Supplementary Materials to better show their differences. This finding establishes that the simulation can systematically capture elasticity, friction, and their interplay.

Rolling robot

With the ultimate goal of simulating soft robots with actuated limbs that engage in inelastic collisions, we first discuss the geometry of rolling robot, then investigate the mechanical response of a single SMA-based actuator. Next, we quantify the locomotion of the robot over three different surfaces: (i) horizontal planar surface, (ii) inclined and declined planar surfaces, and (iii) sinusoidal surfaces.

*** Geometry of rolling robot**

The star-shaped, rolling robot in Fig. 2A is comprised of seven compliant actuators/limbs that are arranged radially. Each limb has a curved part with length \$l_c = 2.2\text{cm}\$ and a straight part

with length $l_s = 0.8\text{cm}$. The natural curvature of the curved part is $\bar{\kappa}_0 \equiv 1/R_c = 120\text{m}^{-1}$. The material density of the rolling robot is $\rho = 1912\text{ kg/m}^3$. The mass center is located at (x_c, y_c) . The height, $H \approx 5\text{ cm}$, is used as the body length. We then discretize the structure into N nodes, shown schematically in Fig. 2A. This corresponds to a degrees of freedom (DOF) vector, $\mathbf{q} = [x_0, y_0, \dots, x_{N-1}, y_{N-1}]^T$, of size $2N$, representing the vertical and horizontal coordinates of each node. Here, the superscript T denotes transposition. The length of each edge – the segment between two consecutive nodes – in this study is $\Delta l \approx 2.5\text{ mm}$, resulting in $N = 84$ nodes (convergence study in the Supplementary Materials).

* Single actuator

Actuation is incorporated into the simulation by varying natural curvature and bending stiffness with time. This variation is measured through characterization of a single SMA-powered actuator, as described next. The electrically-activated SMA wire enables rapid transition between a soft curled unactuated state and a stiff straight-like actuated state^{41,42}. The relative natural curvature $\bar{\kappa}/\bar{\kappa}_0$ and Young's modulus E/E_0 are temperature-dependent, and can change as a function of time during the actuating-cooling process. As shown in Fig. 4A, when SMA is actuated for 0.25s, its natural curvature and Young's modulus increase linearly in a short time period, t_0 , followed by a logistic decay until reaching the unactuated state. We use a piece-wise function to describe the natural curvature of SMA actuators:

$$\bar{\kappa}(t) = \begin{cases} \frac{(n-1)t}{t_0} \bar{\kappa}_0 + \bar{\kappa}_0 & \text{when } t < t_0 \\ \frac{(1-n)}{1+e^{-\tau(t-t_0)}} \bar{\kappa}_0 + n\bar{\kappa}_0 & \text{when } t > t_0, \end{cases} \quad (7)$$

Figure 4: (A) Relative natural curvature $\bar{\kappa}/\bar{\kappa}_0$ and Young's modulus E/E_0 as a function of time during actuating-cooling process of a single SMA actuator. (B) ~~Experimental and simulation data on the location of the robot, represented by the x -coordinate of the centroid of the robot, as a function of time, on horizontal planar surface.~~

where $n = \bar{\kappa}_{\min}/\bar{\kappa}_0$ is the ratio between the minimum curvature (at $t = t_0$) and the initial curvature (at $t = 0$), and τ, \bar{t}, t_0 are numerical parameters obtained from experimental fitting. The change of Young's modulus of SMA follows a similar piecewise piece-wise function. Note that the parameter t_0 is not necessarily equal to the actuation time of 0.25s. As a result, the curvature slightly increases (and Young's modulus decreases) even when the actuator is being heated (at $t_0 < t < 0.25$ s). The primary reason behind behavior is that the fitting function is constrained to be smooth and monotonic (i.e. either increase, decrease, or remain constant). While we could separate the fitting into more than two piece-wise functions involving more fitting parameters, this will lead to added complexity with little improvement in fitting accuracy. With these fitting parameters, we can achieve excellent match between experimental measurements and numerical

simulations performed on a single actuator (details on the fitting can be found in the Supplementary Materials).

* Horizontal planar surface

This is the simplest scenario presented in Fig. 1 (also Movie S1), where the surface normal is ~~perpendicular~~ anti-parallel to gravity. Fig. 4B-5D1 plots the x -coordinate of the centroid of the robot, x_c , with time over four actuation cycles. Note that the different symbols correspond to repeated experimental runs. In our numbering system for the limbs (see Fig. 1), Limb 5 is in contact with the ground at $t = 0$. Upon actuation of Limb 4, the robot rolls to the right and the contact limb changes from 5 to 6. In the next cycle, Limb 5 (the limb to the left of the contact limb) is actuated. We choose the actuation period $\Delta t = 3\text{s}$ (0.25s for actuation and 2.75s for cooling); the single SMA actuator can totally reshape to its original configuration within 3s. Although the simulation ignores the presence of light ultraflexible wires used to connect the robot to a power supply, we find good agreement between experiments and simulations without the aid of fitting parameter.

* Inclined planar surface

Next, we consider planar surfaces that are inclined at an angle θ with respect to the horizontal plane (also see Movie S1). Fig. 5A-B compares the simulation and experimental results for $\theta = +3.0^\circ$, and Fig. 5D1-D2 plots the location of the robot centroid at three different values of θ . We find good agreement between experiments and simulation in all the cases. In particular, we observe

that when the angle of inclination increases from $\theta = -3.0^\circ$ to $\theta = +3.0^\circ$, the distance traveled by the robot decreases in both experiments and simulations. The gait at $\theta = \{-3.0^\circ, +3.0^\circ\}$ is similar to the horizontal planar case described above. Beyond a certain threshold for θ , the robot can no longer move forward due to the increased role of gravity, e.g. the robot fails to roll up the incline when at $\theta = +6.0^\circ$. The simulation also accurately capture this observation.

* Uneven (sinusoidal) surface

We now move to the case of an uneven surface with an outward normal that varies with location. In the previous two cases of planar surfaces that are horizontal or inclined, we could choose the coordinate system such that the surface normal is along y -axis and solve the discrete equations of motion by constraining the degree of freedom corresponding to the motion along y -axis of a contact node. Therefore, the modified mass method employed in our simulation would be unnecessary. However, it is necessary for the more general case of locomotion across unstructured terrain in which the surface normal varies with space. As a representative example shown schematically in Figs. 5B-C, we consider a 3D printed surface that can be described by $f(x) = A \sin(2\pi x/\lambda)$ with amplitude $A = 6.5\text{mm}$ and period $\lambda = 200\text{mm}$. We consider two experimental trials: (i) the robot is initially located at the crest of the surface in Fig. 5B, and (ii) the robot is on the trough in Fig. 5C. Fig. 5D2 shows D5 and D6 show the location of the robot centroid with time from both experiments and simulations. In the *crest* case, the robot rolls once at the first cycle. However, at the second cycle, the robot rolls multiple times, undergoes oscillatory motion, and settles stay at the *trough*. On the other hand, if the locomotion starts with the robot at the trough, the robot

successfully rolls once in the first two cycles, but fails to roll in the third cycle. All of these observations are captured in both experiments and simulations. However, we should also note that our simulator always under-predicts the motion of the rolling robot. We attribute this to the finite thickness of the actuator elements, which is not accounted for in the model

* Computation time

Our novel numerical tool can achieve real-time simulation of the soft rolling robot. In Fig. 6, with a fixed number of vertices, \$N = 84\$, the computation time linearly scales with time step size \$h\$ for all the scenarios. The simulations ran on a single thread of AMD Ryzen 1950X CPU @ 3.4 GHz. Also, our simulator can run faster than real time when the time step size \$h \gtrsim 2.5\text{ms}\$. Numerical issues associated with a large step size appear at \$h \gtrsim 10\text{ms}\$, in which case the computation time is infinite because we cannot get convergence. In summary, the numerical framework can quantitatively predict the motion of soft robots while running faster than real time on one thread of a contemporary desktop processor.

Jumper

Finally, we emphasize the generality of the simulation by examining another soft robot with a different geometry. The SMA-based *jumper* shown in Fig. 7A is an asymmetric circle with radius $\bar{R}_0 \approx 9$ – $\bar{R}_0 \approx 5$ mm. Detailed geometric property of Jumper robot can be found in the Supplementary Materials. When the material is actuated, the whole structure can rise and move forward because of the reaction forces from the ground. To model the tension from the electrical

wire connected at the leading edge of the jumper, we apply a force at the first node; the magnitude and duration of the force are obtained from fitting to experimental data (Supplementary Materials). In Fig. 7A, we show snapshots of the jumper at $t = \{0.000, 0.125, 0.250\}$ s from both experiments and simulations and see qualitative agreement. For quantitative comparison, Fig. 7B ~~presents~~ and C present experimental and simulation data on the normalized position of the first node on the robot as a function of time. The two sets of results – experiments and simulations – appear to be in strong quantitative agreement, providing further evidence for the physical accuracy of our DDG-based formulation. We should also note that, at time $t = 1.5$ s, the x position predicted by numerical simulation has a sudden drop, which is not reflected in the experimental data. The mismatch between numerical simulation and experimental observation is due to the discontinuity in Coulomb’s formulation, i.e. the jumper robot would move only if the external reaction force is out of the frictional cone.

Conclusions

We have introduced a numerical framework for examining the locomotion of limbed soft robots that is adapted from methods popular in the computer graphics community. ~~The simulation seamless integrates~~ Specifically, our numerical framework was developed on the basis of a well-established Discrete Elastic Rods (DER) algorithm. Considering the geometric properties of our soft robotic structures, we simplified the traditional DER method in 2D and ignored the twisting curvature in slender structures, such that the computational efficiency can be significantly improved. To avoid the artificial energy dissipation during the time marching scheme, we replaced the first order,

implicit Euler integration by a second order, symplectic Newmark-beta method, for momentum preservation during the dynamic simulation. For the frictional contact between the rigid wall and the soft material, Coulomb's law was implemented through a modified mass-based method, and this fully implicit framework allows a larger time step for convergence and numerical stability, which is also a prerequisite for real-time simulation. Similarly, the elastic/inelastic collision between the rigid wall and soft robots, related to the rate-dependent viscoelastic behavior of soft material, can be precisely described by the Rayleigh damping matrix. The mechanical response of SMA during actuating-cooling process was first experimentally measured through a single actuator, then fed to the numerical framework to simulate the dynamics of the soft robots. Overall, the simulation can seamless integrate elasticity, actuation, friction, contact, and elastic/inelastic collision to achieve quantitative prediction of the motion of fast moving highly deformable soft robots. FurthermoreImportantly, the simulation can run faster than real-time on a single thread of a contemporary desktop processor. Such computational efficiency makes it ideally suited for algorithms that iterate over a wide variety of parameters in order to select a robot design or locomotion strategy.

Overall, our results show good quantitative agreement between the simulations and experiments, suggesting that our numerical approach represents a promising step towards the ultimate goal of a computational framework for soft robotics engineering. However, further progress depends on additional experimental validation for a wider range of soft robot designs, locomotion gaits, and environmental conditions. The simulation introduced here also needs some prerequisite experimentally measured data, e.g. material properties of soft materials and their mechanical

performance in response to external actuation. It would be meaningful to develop a more general constitutive relations that combines mechanics, electricity, heat, and magnetic field, for the direct simulation of soft robotic dynamics in response to external actuation. Moving forward, it would also be interesting to explore how DDG-based simulation tools that incorporate the formulation presented here can be used to generate optimal locomotion gaits that minimize cost of transport or maximize range for a prescribed energy input.

Methods

Fabrication of Shape Memory Alloy actuators The fabrication process is similar to the one presented in Refs. ^{42,46}. We start the fabrication process by laser-cutting two pieces of thermally conductive tape (H48-2, T-Global) with dimensions of $40 \times 18 \times 0.5$ mm ($55 \times 18 \times 0.5$ mm for the jumping robot) and $80 \times 55 \times 0.5$ mm by a CO₂ laser cutting system (30W VLS 3.50; Universal Laser Systems). Then we apply a layer of elastomer (Ecoflex 00-30, Smooth-On) that is prepared by mixing prepolymer at a 1:1 ratio by mass in a centrifugal mixer (AR-100, THINKY) with a thickness of 0.1mm on top of the small thermally conductive tape and half-cure it in the oven under 50°C for 7 minutes. Next, we place the pre-bent Ω shape SMA wire (0.015 inch diameter, 34×11 mm; Dynalloy) on top of the elastomer and apply another layer of elastomer with a thickness of 0.5mm to encapsulate the SMA wire. Meanwhile, we stretch the larger thermally conductive tape ($80 \times 55 \times 0.5$ mm) to 150% of its original length and apply a 0.1mm thick layer of elastomer on top of it. We place both thermally conductive tape in the oven under 50°C for 7 minutes to half-cure them. After that, we attach the smaller thermally conductive to the middle of the stretched larger

thermally conductive tape, clamp them with binder clips and place the bonded structure back to the oven for 10 minutes to fully cure it. Finally, we cut out the actuator along with the outline of the smaller thermally conductive tape.

Experimental Setup All robots and actuators in the experiment and characterization are powered by a desktop power supply (DIGI360, Electro Industries) under a current of 6A. Each actuator is individually connected to an n-MOSFET (IRL7833, Nfineon for the rolling robot experiment and AO3416, Alpha & Omega Semiconductor for the rest) and the actuation and cooling time are controlled by a single-board microcontroller (Arduino UNO SMD R3, Arduino). The rolling robot and jumper experiment are performed on top of a red linatex sheet (MCM linatex 1.5mm class 2, Linatex Corp of America).

References

1. Shepherd, R. F. *et al.* Multigait soft robot. *Proc. Natl. Acad. Sci.* **108**, 20400–20403 (2011).
2. Seok, S. *et al.* Meshworm: a peristaltic soft robot with antagonistic nickel titanium coil actuators. *IEEE/ASME Trans. Mechatronics* **18**, 1485–1497 (2013).
3. Lin, H.-T., Leisk, G. G. & Trimmer, B. Goqbot: a caterpillar-inspired soft-bodied rolling robot. *Bioinspir. Biomim.* **6**, 026007 (2011).
4. Rich, S. I., Wood, R. J. & Majidi, C. Untethered soft robotics. *Nat. Electron.* **1**, 102 (2018).
5. Buschmann, T. & Trimmer, B. Bio-inspired robot locomotion. *Neurobiology of Motor Control: Fundamental Concepts and New Directions* (2017).

6. Rus, D. & Tolley, M. T. Design, fabrication and control of soft robots. *Nature* **521**, 467–475 (2015).
7. Calisti, M., Corucci, F., Arienti, A. & Laschi, C. Bipedal walking of an octopus-inspired robot. In *Conference on Biomimetic and Biohybrid Systems*, 35–46 (Springer, 2014).
8. Onal, C. D. & Rus, D. Autonomous undulatory serpentine locomotion utilizing body dynamics of a fluidic soft robot. *Bioinspir. Biomim.* **8**, 026003 (2013).
9. Suzumori, K., Endo, S., Kanda, T., Kato, N. & Suzuki, H. A bending pneumatic rubber actuator realizing soft-bodied manta swimming robot. In *IEEE Int. Conf. Robot. Autom.*, 4975–4980 (IEEE, 2007).
10. Marchese, A. D., Onal, C. D. & Rus, D. Autonomous soft robotic fish capable of escape maneuvers using fluidic elastomer actuators. *Soft Robot.* **1**, 75–87 (2014).
11. Fei, Y. & Xu, H. Modeling and motion control of a soft robot. *IEEE Transactions on Industrial Electronics* **64**, 1737–1742 (2016).
12. Coevoet, E. *et al.* Software toolkit for modeling, simulation, and control of soft robots. *Advanced Robotics* **31**, 1208–1224 (2017).
13. Duriez, C. Control of elastic soft robots based on real-time finite element method. In *2013 IEEE International Conference on Robotics and Automation*, 3982–3987 (IEEE, 2013).
14. Runge, G. & Raatz, A. A framework for the automated design and modelling of soft robotic systems. *CIRP Annals* **66**, 9–12 (2017).

15. Goury, O. & Duriez, C. Fast, generic, and reliable control and simulation of soft robots using model order reduction. *IEEE Transactions on Robotics* 1–12 (2018).
16. Chenevier, J., González, D., Aguado, J. V., Chinesta, F. & Cueto, E. Reduced-order modeling of soft robots. *PloS one* **13**, e0192052 (2018).
17. Hiller, J. & Lipson, H. Dynamic simulation of soft multimaterial 3d-printed objects. *Soft robotics* **1**, 88–101 (2014).
18. Cheney, N., Bongard, J. & Lipson, H. Evolving soft robots in tight spaces. In *Proceedings of the 2015 annual conference on Genetic and Evolutionary Computation*, 935–942 (ACM, 2015).
19. Zhou, X., Majidi, C. & O'Reilly, O. M. Soft hands: An analysis of some gripping mechanisms in soft robot design. *International Journal of Solids and Structures* **64**, 155–165 (2015).
20. Grazioso, S., Di Gironimo, G. & Siciliano, B. A geometrically exact model for soft continuum robots: The finite element deformation space formulation. *Soft robotics* (2018).
21. Renda, F. *et al.* A unified multi-soft-body dynamic model for underwater soft robots. *The International Journal of Robotics Research* **37**, 648–666 (2018).
22. Grinspun, E., Desbrun, M., Polthier, K., Schröder, P. & Stern, A. Discrete differential geometry: an applied introduction. *ACM SIGGRAPH Course* **7**, 1–139 (2006).

23. Kaufman, D. M., Tamstorf, R., Smith, B., Aubry, J.-M. & Grinspun, E. Adaptive nonlinearity for collisions in complex rod assemblies. *ACM Transactions on Graphics (TOG)* **33**, 123 (2014).
24. de Payrebrune, K. M. & O'Reilly, O. M. On constitutive relations for a rod-based model of a pneu-net bending actuator. *Extreme Mechanics Letters* **8**, 38–46 (2016).
25. de Payrebrune, K. M. & O'Reilly, O. M. On the development of rod-based models for pneumatically actuated soft robot arms: A five-parameter constitutive relation. *Int. J. Solids Struct.* (2017).
26. Trivedi, D., Lotfi, A. & Rahn, C. D. Geometrically exact models for soft robotic manipulators. *IEEE Trans. Robot.* **24**, 773–780 (2008).
27. Rucker, D. C., Jones, B. A. & Webster III, R. J. A geometrically exact model for externally loaded concentric-tube continuum robots. *IEEE Trans. Robot.* **26**, 769–780 (2010).
28. Bern, J. M., Kumagai, G. & Coros, S. Fabrication, modeling, and control of plush robots. In *2017 IEEE/RSJ International Conference on Intelligent Robots and Systems (IROS)*, 3739–3746 (IEEE, 2017).
29. Jawed, M. K., Novelia, A. & O'Reilly, O. *A primer on the kinematics of Discrete Elastic Rods*. SpringerBriefs in Applied Sciences and Technology (Springer-Verlag, 2018).
30. Bergou, M., Audoly, B., Vouga, E., Wardetzky, M. & Grinspun, E. Discrete viscous threads. In *ACM Trans. Graph.*, vol. 29, 116 (ACM, 2010).

31. Bergou, M., Wardetzky, M., Robinson, S., Audoly, B. & Grinspun, E. Discrete elastic rods. In *ACM Trans. Graph.*, vol. 27, 63 (ACM, 2008).
32. Shen, Z., Huang, J., Chen, W. & Bao, H. Geometrically exact simulation of inextensible ribbon. In *Comput. Graph. Forum*, vol. 34, 145–154 (Wiley Online Library, 2015).
33. Baraff, D. & Witkin, A. Large steps in cloth simulation. In *ACM Trans. Graph.*, 43–54 (ACM, 1998).
34. Grinspun, E., Hirani, A. N., Desbrun, M. & Schröder, P. Discrete shells. In *Symposium on Computer Animation*, 62–67 (Eurographics Association, 2003).
35. Audoly, B. *et al.* A discrete geometric approach for simulating the dynamics of thin viscous threads. *J. Comput. Phys.* **253**, 18–49 (2013).
36. Batty, C., Uribe, A., Audoly, B. & Grinspun, E. Discrete viscous sheets. In *ACM Trans. Graph.*, vol. 31, 113 (ACM, 2012).
37. Goldberg, N. N. *et al.* On planar discrete elastic rod models for the locomotion of soft robots. *Soft robotics* (2019).
38. O’Reilly, O. M. *Modeling Nonlinear Problems in the Mechanics of Strings and Rods* (Springer, 2017).
39. Chen, D., Levin, D. I., Matusik, W. & Kaufman, D. M. Dynamics-aware numerical coarsening for fabrication design. *ACM Transactions on Graphics (TOG)* **36**, 84 (2017).

40. Raux, P., Reis, P. M., Bush, J. & Clanet, C. Rolling ribbons. *Physical review letters* **105**, 044301 (2010).
41. Huang, X. *et al.* Chasing biomimetic locomotion speeds: Creating untethered soft robots with shape memory alloy actuators. *Science Robotics* **3**, eaau7557 (2018).
42. Huang, X. *et al.* Highly dynamic shape memory alloy actuator for fast moving soft robots. *Advanced Materials Technologies* 1800540 (2019).
43. Bertails-Descoubes, F., Cadoux, F., Daviet, G. & Acary, V. A nonsmooth newton solver for capturing exact coulomb friction in fiber assemblies. *ACM Transactions on Graphics (TOG)* **30**, 6 (2011).
44. Kane, C., Marsden, J. E., Ortiz, M. & West, M. Variational integrators and the newmark algorithm for conservative and dissipative mechanical systems. *International Journal for Numerical Methods in Engineering* **49**, 1295–1325 (2000).
45. Huang, W. & Jawed, M. K. Newmark-beta method in discrete elastic rods algorithm to avoid energy dissipation. *Journal of Applied Mechanics* **86**, 084501 (2019).
46. Huang, X., Kumar, K., Jawed, M., Ye, Z. & Majidi, C. Soft electrically actuated quadruped (seq)-integrating a flex circuit board and elastomeric limbs for versatile mobility. *IEEE Robotics and Automation Letters* (2019).

Supplementary information Supplementary information is available for this paper.

Acknowledgements W.H. and M.K.J. acknowledge support from the Henry Samueli School of Engineering and Applied Science, University of California, Los Angeles. X.H. and C.M. acknowledge support from the Army Research Office (Grant #: W911NF-16-1-0148; Dr. Samuel Stanton).

Competing Interests The authors declare that they have no competing financial interests.

Correspondence Correspondence and requests for materials should be addressed to C.M. and M.K.J. (email: cmajidi@andrew.cmu.edu and khalidjm@seas.ucla.edu).

Figure 5: Snapshots of rolling robot locomotion on (A) inclined surface ($\theta = +3.0^\circ$), (B) crest of sinusoidal surface, and (C) trough of sinusoidal surface, from simulations and experiments. See Movie S1 for videos comparing experiments and simulations. Normalized x -coordinate of the robot centroid with time from experiments (symbols) and simulations (solid lines) for (D1) planar surfaces, \$\theta = 0.0^\circ\$ ; (D2) inclined surfaces, \$\theta = +3.0^\circ\$ ; (D3) declined surfaces, \$\theta = -3.0^\circ\$ ; (D4) inclined surfaces, \$\theta = +6.0^\circ\$ ; (D5) sinusoidal surfaces, crest; and (D6) sinusoidal surfaces, trough.

Figure 6: The ratio between computational time and wall-clock time as a function of time step size h in different scenarios with $N = 84$.

Figure 7: (A) Snapshots of the *jumper* from experiments (left) and simulations (right) at different time steps (Movie S1). Normalized (B) Normalized x position and (C) y position of the first node (marked in A) in jumper robot as a function of time from experiments (symbols) and simulations (solid lines).

Supplementary Materials for Dynamic Simulation of Articulated Soft Robots

Weicheng Huang^{1†}, Xiaonan Huang^{2†}, Carmel Majidi^{2*}, and M. Khalid Jawed^{1*}

¹*Department of Mechanical and Aerospace Engineering, University of California, Los Angeles,
420 Westwood Plaza, Los Angeles, CA 90095*

²*Department of Mechanical Engineering, Carnegie Mellon University,
5000 Forbes Avenue, Pittsburgh, PA 15213*

† *W.H. and X.H. contributed equally to this work.*

**To whom correspondence should be addressed:*

cmajidi@andrew.cmu.edu and khalidjm@seas.ucla.edu.

Supplementary Methods

* Actuator characterization

In this section, we study the mechanical property of a single Shape Memory Alloy (SMA) actuator. We manufactured a single curved SMA actuator sample with the following geometric parameters: arc length $L_s = 33\text{mm}$, undeformed curvature $\bar{\kappa}_0 \equiv 1/R_0 = 97.03\text{m}^{-1}$, thickness $h = 1.72\text{mm}$, width $w = 19.42\text{mm}$. The material density is $\rho = 1920\text{kg/m}^3$, which is similar to our previous study¹⁻³.

Figure S1: Experiments of deformed SMA actuator in (A) unactuated state and (B) actuated state under loading. Simulation results of initial configuration (dashed line) and deformed configuration (solid line) in (C) unactuated state and (D) actuated state.

We employ load-displacement relation to find the Young's modulus of a single SMA actuator for both unactuated and actuated state. In Fig. S1(A) (unactuated state) and (B) (actuated state), we use a 20g weight to apply a vertical load at the end of the actuator and evaluate the end displacement. Then we use simulation to find the Young's modulus that best matches the experimental results. In Fig. S1(C) and (D), we plot the best fit configurations of SMA actuator before/after loading. Here, the Young's modulus we found in unactuated state is $E_0 = 3.0$ MPa, and $E_{\max} = 8.04$ MPa in actuated state; these are comparable to the values reported in Ref. ³.

Next, we model the dynamics of a single SMA-based actuator during actuating/cooling pro-

cess. In Fig. S2(A) and (B), we show the undeformed shape and maximum response shape of SMA-based actuator during heating-cooling process separately. The heating time used in this experiment is 0.25s, and cooling time is 2.75s. We assume that the natural curvature $\bar{\kappa}(t)$ follows a piece-wise function,

$$\bar{\kappa}(t) = \begin{cases} \frac{(n_1-1)t}{t_0} \bar{\kappa}_0 + \bar{\kappa}_0 & \text{when } t < t_0 \\ \frac{(1-n_1)}{1+e^{-\tau(t-\bar{t})}} \bar{\kappa}_0 + n_1 \bar{\kappa}_0 & \text{when } t > t_0, \end{cases} \quad (1)$$

and similarly for Young's modulus $E(t)$,

$$E(t) = \begin{cases} \frac{(n_2-1)t}{t_0} E_0 + E_0 & \text{when } t < t_0 \\ \frac{(1-n_2)}{1+e^{-\tau(t-\bar{t})}} E_0 + n_2 E_0 & \text{when } t > t_0, \end{cases} \quad (2)$$

where n_1 and n_2 are the ratios between unactuated state and actuated state, $n_1 = \bar{\kappa}_{\min}/\bar{\kappa}_0$, $n_2 = E_{\max}/E_0$ (the actuated curvature is smaller than unactuated state, while Young's modulus follows an opposite pattern), $t_0 = 0.05\text{s}$, $\bar{t} = 1.4\text{s}$, and $\tau = 3.4\text{s}^{-1}$ are from experimental fitting.

The ratio of Young's modulus, $n_2 = E_{\max}/E_0 = 2.68$, can be easily obtained based on previous loading experiments; another experimentally evaluated parameter, minimum natural curvature, is $\bar{\kappa}_{\min} = 20\text{m}^{-1}$, resulting in $n_1 = 0.21$. ~~In Fig. S2(C), we plot the relative Young's modulus E/E_0 and relative natural curvature $\bar{\kappa}/\bar{\kappa}_0$ as a function of time during actuating process; we~~ We use these fitting parameters to perform this dynamic process in our simulation, and plot the relative beam end position, X_{end}/\bar{R}_0 and Y_{end}/\bar{R}_0 , as a function of time during this dynamic process in Fig. S2(D). ~~Here we can find a good match between experimental data and simulation results~~ Then, in Fig. S2(D), we plot these best fitting parameters, e.g. relative Young's modulus E/E_0 and relative natural curvature $\bar{\kappa}/\bar{\kappa}_0$, as a function of time, during the actuating process.

* Inelastic collision

The collision between soft material and rigid platform usually results in partly inelastic collision⁴. Consider the scenario in Fig. S3(A) where the rolling robot is dropped from an initial height, L_0 . When its two limbs touch the ground, we constraint these two nodes and manually set their normal velocities (relative to the ground) as zeros, to perform the collision. However, as the other parts of rolling robot still have non-zero velocities, the whole structure will be compressed and the kinetic energy will transfer into elastic potential energy. If the viscoelastic behavior pertinent to collision is not considered, the kinetic energy of rolling robot will transfer into potential energy and the structure will rebound to a certain height close to its initial dropped height. Without viscoelasticity, this rebound height is not controllable based on material property.

We add a damping force $\mathbf{F}^d = -(\alpha\mathbb{M} + \beta\mathbb{K})\mathbf{v}$ to the system to model the viscoelastic behavior and inelastic collision of SMA, where $\alpha, \beta \in \mathbb{R}^+$, \mathbb{M} is the mass matrix, and $\mathbb{K} = -\frac{\partial}{\partial \mathbf{q}}(\mathbf{F}^s + \mathbf{F}^b)$ is the tangent stiffness matrix⁴. The viscosity parameter α is related to external environment damping and can cause momentum dissipation during rigid body motion. As our robot is moving in the air, the effect of environmental damping is negligible, such that $\alpha = 0$. The parameter β is related to material damping and causes energy dissipation only during the deformed process. For rigid body motion, the internal elastic force, $\mathbf{F}^s + \mathbf{F}^b = \mathbb{K}\mathbf{q}$, is zero; its time derivative (and, therefore, the internal damping force, $\mathbf{F}^d = -\beta\mathbb{K}\mathbf{v}$), should also be zero.

In Fig. S3(B), we plot the normalized y -coordinate of the centroid of the robot, y_c/H , as a function of time, at different values of β . As expected, we find that the rebound height – local

maxima in the time series of y_c – decreases with increasing value of β . When $\beta = 1e^{-2}$, there is almost no rebound (i.e. perfectly sticky surface) when the rolling robot is dropped from the height of its own size. We choose $\beta = 3e^{-3}$ that best matches the experimental data. This value is used in all of our simulations on the rolling and jumper robots.

* Rolling ribbon

In the numerical study of rolling ribbon, the arc length we chose for the circular ribbon is $L_0 = 0.3\text{m}$, resulting in $\bar{R}_0 = 0.3/2\pi \approx 0.048\text{m}$. The ribbon thickness is $r_0 = 1\text{mm}$, poisson ratio $\nu = 0.5$ (incompressible material), material density $\rho = 1237.52\text{kg/m}^3$, and we vary the Young's modulus, E , from 1 MPa to 100 MPa to vary the governing dimensionless group Γ_g . The damping parameters are $\alpha = 0$ and $\beta = 1e^{-3}$; however, since we do not quantitatively study the transient dynamics, these parameters do not affect the final deformed configuration and motion patterns.

Here, we show the different motion patterns of rolling ribbon. Consider a circular ribbon with $\Gamma_g = 0.57^5$ that is moving in a declined surface with $\theta = -17.19^\circ$. Its deformed configuration is shown in Fig. S4(A). The displacement of one boundary node along the declined surface, defined as t_b , is used to quantify the different motion patterns of rolling ribbon. When the relative frictional coefficient $\mu/\tan\theta = 0$, the ribbon slides along the declined surface, without any rotation, resulting in a smooth curve in Fig. S4(B). If the relative frictional coefficient is non zero but smaller than 1, we get a motion that combines sliding and rotating; the zigzag nature of the displacement vs. time data shown in Fig. S4(C) arises from the following: the node periodically comes in contact with the surface and, as a result, its velocity slows down. In Fig. S4(D), we show

the tangential displacement of boundary node as a function of time when the ribbon is rotating along the declined surface, without any sliding. From the figure, we can clearly see that the boundary node is totally fixed with the declined surface without any relative displacement once the node touches the ground.

Note that when the ribbon is sliding (pure sliding and the combination of sliding and rotating) along the declined surface, its velocity will continue to increase if environmental damping $\alpha = 0$. In contrast, the speed of pure rotating case remains fixed even when no external damping force is applied into the system.

* Measurement of Coefficients of Friction

The dynamic coefficient of friction μ between the thermally conductive tape and rubber is characterized by giving a slight push to the robot and recording the angle of inclination, θ_0 , where it starts to slip down at a steady speed; this method is inspired by Ref. ³. The coefficient of friction is prescribed by the relation $\mu = \tan \theta_0$. We also assume that the static frictional coefficient is same as the dynamic coefficient.

* Geometry of Jumper Robot

The geometry of Jumper Robot is an asymmetric circle, as shown in Fig. S5A. The undeformed radius is $\bar{R}_0 \approx 0.050\text{m}$, and other two geometric parameters, $\theta_1 \approx 40^\circ$ and $\theta_2 \approx 60^\circ$, are fitted from experimental image. The material used in Jumper Robot is identical to the one used in

the rolling robot, discussed in the previous section.

Also, as shown in Fig. S5B, we applied a constant force \mathbf{F}_0 on the first node on jumper robot over $t_0 \leq t \leq t_1$ (t_0, t_1 are parameters obtained from data fitting) to emulate the presence of the electrical wire. After fitting to experimental data, the horizontal and vertical components of this external force are $F_0^x = 0.05Mg$ and $F_0^y = 0.8Mg$, where M is the total mass of jumper robot. This force is applied at $t_0 = 0.1s$, and ended at $t_1 = 0.15s$.

* Computational time

~~Our novel numerical tool can achieve real-time simulation of soft rolling robot. In Fig. ??, with a fixed number of vertices, $nv = 84$, the computational time linearly scales with time step size h for all the scenarios. The simulations ran on a single thread of AMD Ryzen 1950X CPU @ 3.4 GHz. Also, our simulator can run faster than real time when time step size $h \gtrsim 2.5ms$. Numerical issues associated with large step size appears at $h \gtrsim 10ms$. In summary, the numerical framework can quantitatively predict the motion of soft robots while running faster than real time on one thread of a contemporary desktop processor.~~

~~Convergence study of rolling robot for (A) time discretization (with $nv = 84$ fixed) and (B) space discretization (with $h = 0.1ms$ fixed). From (1)–(4): convergence study of (1) planar case; (2) inclined surface with $\theta = +3.0^\circ$; (3) sinusoidal surface with initial location of the robot at crest, and (4) sinusoidal surface with initial location at trough.~~

* Convergence study

Our simulation is robust and shows good convergence with both time and space discretization. In Fig. S6(A), we plot the normalized centroid of rolling robot x_c/H , as a function of time, at a fixed number of vertices ~~$nv = 84$~~ $N = 84$ and different values of time step size h . Our simulations show good convergence with time for all the following cases: (A1) planar motion, (A2) inclined surface with $\theta = +3.0^\circ$, (A3) crest, and (A4) trough. Numerical issues begin to appear beyond $h = 10$ ms. Similarly, we vary number of vertices, ~~nv~~ N , in Fig. S6(B) and fix the time step size at $h = 0.1$ ms to show the convergence with space discretization of our simulator. The simulation fails to make quantitative prediction around ~~$nv \approx 50$~~ $N \approx 50$, e.g. see the data corresponding to ~~$nv = 49$~~ $N = 49$ in Fig. S6(B4). Similar data can be found in Table 1 and Table 2.

Supplementary Video

Movies S1 (.mp4 format): Comparison between experiments and simulations.

Supplementary References

1. Huang, X. *et al.* Chasing biomimetic locomotion speeds: Creating untethered soft robots with shape memory alloy actuators. *Science Robotics* **3**, eaau7557 (2018).
2. Huang, X. *et al.* Highly dynamic shape memory alloy actuator for fast moving soft robots. *Advanced Materials Technologies* 1800540 (2019).

Table 1: Final normalized centroid x coordinate, x_c/H .

time step size h	planar, $\theta = 0.0^\circ$	inclined, $\theta = +3.0^\circ$	Sin curve, crest	Sin curve, trough
$h = 5\text{ms}$	1.1660	1.0958	2.0012	0.3206
$h = 1\text{ms}$	1.1347	1.0975	1.9549	0.3168
$h = 0.5\text{ms}$	1.1356	1.1037	1.9577	0.3122
$h = 0.1\text{ms}$	1.1327	1.1050	1.9688	0.3025
$h = 0.05\text{ms}$	1.1373	1.0993	1.9672	0.3096

Table 2: Final normalized centroid x coordinate, x_c/H .

Number of vertices	planar, $\theta = 0.0^\circ$	inclined, $\theta = +3.0^\circ$	Sin curve, crest	Sin curve, trough
$N = 49$	1.1055	1.0720	1.9289	0.0444
$N = 84$	1.1327	1.1050	1.9688	0.3025
$N = 98$	1.1356	1.0810	1.9598	0.3033
$N = 105$	1.1528	1.0872	1.9740	0.2945
$N = 112$	1.1573	1.1085	1.9738	0.3035

3. Goldberg, N. N. *et al.* On planar discrete elastic rod models for the locomotion of soft robots. *Soft robotics* (2019).
4. Chen, D., Levin, D. I., Matusik, W. & Kaufman, D. M. Dynamics-aware numerical coarsening for fabrication design. *ACM Transactions on Graphics (TOG)* **36**, 84 (2017).
5. Raux, P., Reis, P. M., Bush, J. & Clanet, C. Rolling ribbons. *Physical review letters* **105**, 044301 (2010).

Figure S2: (A) The shape of SMA-based actuator before actuating. (B) Maximum response of SMA-based actuator during actuating process. (C) Relative beam end displacement X_{end}/\bar{R}_0 and Y_{end}/\bar{R}_0 from both experiments and simulations. (D) Relative Young's modulus E/E_0 (red) and relative natural curvature $\bar{\kappa}/\bar{\kappa}_0$ (blue) change as a function of time during actuating/cooling process. ~~(D) Relative beam end displacement X_{end}/\bar{R}_0 and Y_{end}/\bar{R}_0 from both experiments and simulations.~~

Figure S3: (A) Collision process between rolling robot and rigid platform. (B) normalized center position y_c/H as a function of time at different values of material damping parameter, β (solid lines) and experimental data (symbols).

Figure S4: (A) Configuration of rolling ribbon with $\Gamma_g = 0.57$. Relative boundary displacement t_b/R_0 as function of time for different motion patterns: (B) pure sliding, (C) combination of sliding and rotating, and (D) pure rotating.

Figure S5: (A) Geometry of jumper robot. (B) Illustration of the drag Force from electrical wire.

Figure S6: Convergence study of rolling robot for (A) time discretization (with $N = 84$ fixed) and (B) space discretization (with $h = 0.1\text{ms}$ fixed). From (1) – (4): convergence study of (1) planar case; (2) inclined surface with $\theta = +3.0^\circ$; (3) sinusoidal surface with initial location of the robot at crest, and (4) sinusoidal surface with initial location at trough.

Reviewers' Comments:

Reviewer #1:

Remarks to the Author:

The authors have responded appropriately to the reviewer's comments and have supplemented and amended their manuscript accordingly.

Minor remarks:

- Fig 4.: Please mention that the figure shows the fitted curves and not the actual behaviour.
- There is a reference to figure 4a, although figure 4 does not contain a and b anymore.

Reviewer #2:

Remarks to the Author:

The authors of the article made a very complete response to my comments and to the comments of other reviewers. I think their work to correct the article is very good.

In addition, the comments provided allow a better understanding of the novelty of this study.

References to the state of the art were added to the paper, showing that this work builds on an important existing base and that there is little novelty on the modeling aspects, but that the novelty is more on the application.

I still have a question about the used numerical scheme (implicit Newmark-beta) which is combined with the modified mass method. In [33] where this method is used, a time-stepping type method is used with a low order dissipative scheme, which is not the case in this paper.

In the literature of "non-smooth mechanics" which is interested precisely in these problems of contact in dynamics, it is known that there are two types of approach:

- the "event-driven" which stops integration at the time of contact and allows the use of high order integration methods to better conserve energy on contact or,
- "time-stepping" where all the contacts appearing in a time step are processed at the same time as impulses. Then weak order schemes are often used (often dissipative in energy).

I understand the interest of the scheme used here: conserving energy out of contact ... but what is its behavior during contact?

In particular, how does the integration scheme behave during sudden changes in the speed vector at the time of contact?

In the paper it is mentioned that the velocity and position are modified to enforce de contact constraint... but how it influences the energy ?

If we completely remove the Rayleigh damping we should obtain an infinite rebound without loss / creation of energy? is it the case ?

To my opinion Rayleigh damping has also a numerical effect here and/or the scheme is modified at contact time.

Reviewer #3:

Remarks to the Author:

Thanks to the authors for their careful revisions. The additional detail and new organization make the manuscript more clear. I have no further questions or concerns.

Response to the Reviewers' comments for manuscript #:NCOMMS-19-539396
“Dynamic Simulation of Articulated Soft Robots ”
W. Huang, X. Huang, C. Majidi, and M. K. Jawed

Here, we provide a summary of the changes made to the revised manuscript, followed by a detailed response to each of the Reviewers' comments/suggestions (**in bold**). Please refer to the “diff. version” attached at the end of this document for a detailed account of all the changes, corrections, and additions to the revised versions of the manuscript. In that document, the original text that was modified/deleted is struck-through (in black) and the revised/new text is in blue.

1. Summary of changes made to the manuscript:

- We modified the format based on the publisher's suggestions.
- We clearly mentioned the curves in Fig. 4 are from fitting at the “Single actuator” subsection.
- We reduced the size of the text by ~400 words (based on the guidance from the editorial office) and fixed minor typographical errors and implemented stylistic corrections of our own.

2. Response to Reviewer I

Comment 1. The authors have responded appropriately to the reviewer's comments and have supplemented and amended their manuscript accordingly.

We thank the reviewer for his/her recommendation.

Comment 2. Minor remarks:

- **Fig 4.:** Please mention that the figure shows the fitted curves and not the actual behaviour.
- **There is a reference to figure 4a, although figure 4 does not contain a and b anymore.**

We thank the review for his/her careful concern. We did the following modifications based on review's suggestions:

We added a sentence on page 15 in our description of a single actuator,

“Notice that the plot here is from experimental fitting, see Supplementary Figure S2 for details”

We also deleted “A” when referring to Fig. 4 on page 15,

“As shown in Fig. 4,”

3. Response to Reviewer II

Comment 1. The authors of the article made a very complete response to my comments and to the comments of other reviewers. I think their work to correct the article is very good.

In addition, the comments provided allow a better understanding of the novelty of this study. References to the state of the art were added to the paper, showing that this work builds on an important existing base and that there is little novelty on the modeling aspects, but that the novelty is more on the application.

We thank the reviewer for his/her positive feedback.

Comment 2. I still have a question about the used numerical scheme (implicit Newmark-beta) which is combined with the modified mass method. In [33] where this method is used, a time-stepping type method is used with a low order dissipative scheme, which is not the case in this paper.

In Ref. [33] (*Baraff and Witkin. "Large steps in cloth simulation." Proceedings of the 25th annual conference on Computer graphics and interactive techniques. 1998*), they mainly focused on the static behavior of thin elastic objects in contact with rigid bodies, e.g. cloth dropping onto a table. However, in the dynamics of the soft robot, the inertial force is also important during its locomotion. We used a second order time marching scheme that is momentum preserving, see Ref. [*Huang, Weicheng, and Mohammad Khalid Jawed. "Newmark-Beta Method in Discrete Elastic Rods Algorithm to Avoid Energy Dissipation." Journal of Applied Mechanics 86.8 (2019).*] for a detailed discussion of the Newmark-beta time marching scheme in DER method. Also see our response to the next comment of the Reviewer.

Comment 3. In the literature of “non-smooth mechanics” which is interested precisely in these problems of contact in dynamics, it is known that there are two types of approach:

- the “event-driven” which stops integration at the time of contact and allows the use of high order integration methods to better conserve energy on contact or,
- “time-stepping” where all the contacts appearing in a time step are processed at the same time as impulses. Then weak order schemes are often used (often dissipative in energy).

I understand the interest of the scheme used here: conserving energy out of contact ... but what is its behavior during contact? In particular, how does the integration scheme behave during sudden changes in the speed vector at the time of contact?

Our methods employ the latter technique (“time-stepping”) and process all the contacts at the same time. In this case, if the time step size is small enough, the energy dissipation is often negligible. Since our physical system (soft robots) includes damping, we find our scheme to show very good convergence behavior (with time step size). Moreover, our method can be easily modified to include adaptive time step size (“event-driven” where integration stops when contact is detected) and achieve even better convergence with time step size.

In the revised Supplementary Information, we have added a new figure (Supplementary Figure 5) and the following discussion to Supplementary Methods:

“Referring to Supplementary Figure 5, we also note that the convergence with time step size improves when material damping (non-zero β) is introduced. In our simulation scheme, we keep a fixed value of time step size, h . When a node touches the ground, we do not stop integrating the equations of motion; rather, we fix the appropriate degrees of freedom. This introduces some dependence of the simulation result on time step size especially at zero material damping. Our simulation method can be easily extended to use adaptive time step size for better convergence behavior.”

Comment 4. In the paper it is mentioned that the velocity and position are modified to enforce the contact constraint... but how does it influence the energy? If we completely remove the Rayleigh damping we should obtain an infinite rebound without loss/creation of energy? is it the case? In my opinion Rayleigh damping also has a numerical effect here and/or the scheme is modified at contact time.

This dissipates energy and the amount of dissipation is dependent on the size of the time step. If there is no dissipation of energy (and no material damping), the rebound height should be equal to the initial height. Referring to Supplementary Figure 5(a), we note that this is not the case; the rebound height decreases with each “bounce” from the ground. The rebound height varies with the size of the time step. However, we notice in Supplementary Figure 5(b) that introduction of material damping removes the dependence on time step size. Since our soft robots include material damping, we are satisfied by the convergence of the simulation with time step size. Also see our response to Comment 3 of Reviewer II that explains how our simulation method can be modified to “fix” this issue.

4. Response to Reviewer III

Comment 1. Thanks to the authors for their careful revisions. The additional detail and new organization make the manuscript more clear. I have no further questions or concerns.

We thank the reviewer for his/her valuable time, consideration, and positive assessment.